# Single-molecule FRET reveals multiscale chromatin dynamics modulated by HP1α

Sinan Kilic [1,4], Suren Felekyan[2], Olga Doroshenko[2], Iuliia Boichenko[1], Mykola Dimura[2], Hayk Vardanyan[2], Louise C. Bryan[1], Gaurav Arya[3], Claus A.M. Seidel [2] & Beat Fierz [1]

The dynamic architecture of chromatin fibers, a key determinant of genome regulation, is poorly understood. Here, we employ multimodal single-molecule Förster resonance energy transfer studies to reveal structural states and their interconversion kinetics in chromatin fibers. We show that nucleosomes engage in short-lived (micro- to milliseconds) stacking interactions with one of their neighbors. This results in discrete tetranucleosome units with distinct interaction registers that interconvert within hundreds of milliseconds. Additionally, we find that dynamic chromatin architecture is modulated by the multivalent architectural protein heterochromatin protein 1α (HP1α), which engages methylated histone tails and thereby transiently stabilizes stacked nucleosomes. This compacted state nevertheless remains dynamic, exhibiting fluctuations on the timescale of HP1α residence times. Overall, this study reveals that exposure of internal DNA sites and nucleosome surfaces in chromatin fibers is governed by an intrinsic dynamic hierarchy from micro- to milliseconds, allowing the gene regulation machinery to access compact chromatin.

[1] Laboratory of Biophysical Chemistry of Macromolecules, Institute of Chemical Sciences and Engineering (ISIC), Ecole Polytechnique Fédérale de Lausanne (EPFL), 1015 Lausanne, Switzerland. [2] Institut für Physikalische Chemie, Lehrstuhl für Molekulare Physikalische Chemie, Heinrich-Heine-Universität, Universitätsstraße 1, 40225 Düsseldorf, Germany. [3] Pratt School of Engineering, Duke University, 144 Hudson Hall, Box 90300, Durham, NC 27708, USA. [4] Present address: Department of Molecular Mechanisms of Disease, University of Zurich, 8057 Zurich, Switzerland. Correspondence and requests for materials should be addressed to C.A.M.S. (email: cseidel@hhu.de) or to B.F. (email: beat.fierz@epfl.ch)

Chromatin is critical to gene regulatory processes, as it dictates the accessibility of DNA to proteins such as transcription factors (TFs) and gene expression machinery[1]. The elucidation of the structure and dynamics of chromatin is a challenge spanning orders of magnitude in spatial (Å to micrometers) and temporal (sub-microseconds to hours) scales[2]. Genomic approaches have enabled researchers to probe the structure of chromatin in vivo[3–5], albeit as static snapshots and averaged over cellular populations. High-resolution structural studies on reconstituted chromatin provided models of chromatin as a two-start helix with two intertwined stacks of nucleosomes and compact tetranucleosomes as basic units (Fig. 1a)[6,7]. Within such a two-start fiber context, inter-nucleosome interactions are mediated by the H4 tail contacting the H2A acidic patch[1], and by a contact between the C-terminal helices of H2A and H2B[6,7]. Other experiments have supported solenoid chromatin structural models[8] or mixed, heterogeneous populations[9], depending on linker DNA length and the presence of linker histones. As observed in the cryo-EM structure of a chromatin fiber (Fig. 1a), tetranucleosomes arrange in a defined interaction register (i.e., defining which nucleosomes interact with each other).

Irrespective of the local architecture, chromatin structure is highly dynamic: Mononucleosomes exhibit partial unwrapping of nucleosome-wound DNA[10–13], which modulates binding site accessibility for TFs[14,15] and controls the rate of transcription by RNA polymerase[16]. Dynamic rearrangements beyond the nucleosome were observed using fluorescence approaches in tri-nucleosomes[17] and using force spectroscopy on chromatin fibers under tension[18–21]. However, structural rearrangements in unperturbed chromatin fibers, and the timescales thereof, remain unresolved.

Heterochromatin protein 1α (HP1α, CBX5), a defining component of transcriptionally silent heterochromatin, has been shown to interact with H3 tri-methylated at lysine 9 (H3K9me3)

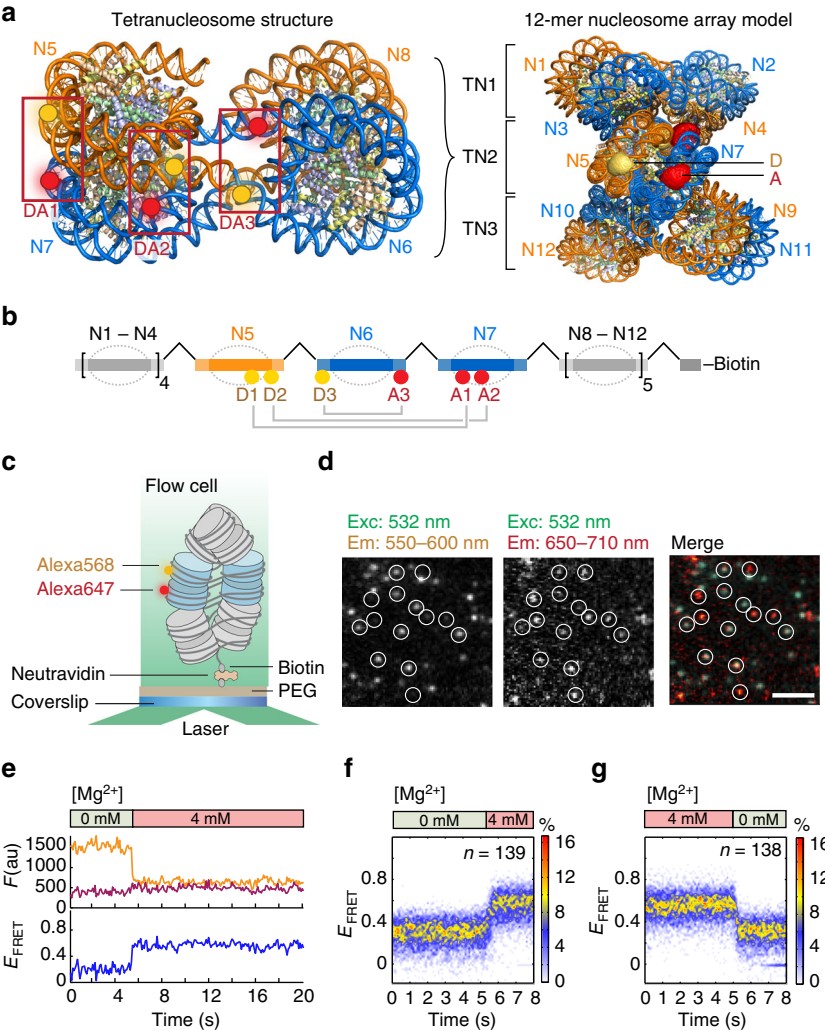

**Fig. 1** smFRET system to detect real-time chromatin conformational dynamics. **a** Left: Tetranucleosome structure based on ref. [6] showing the three dye pairs DA1, DA2, and DA3. Right: 12-mer chromatin fiber as a stack of three tetranucleosome (TN) units, modeled using the cryo-EM structure of a chromatin fiber[7]. For exact dye positions, see Supplementary Fig. 1. The middle tetranucleosome carries the fluorescent labels, whose accessible volume is displayed. D donor, A acceptor labels, N nucleosomes. **b** Schematic view of the preparative DNA ligation used to introduce fluorescent labels. **c** Scheme of the TIRF experiment to measure intra-array smFRET. **d** Microscopic images showing FRET data of single chromatin arrays at 4 mM Mg²⁺, scale bar: 5 μm. **e** Trace from dynamic compaction of chromatin fibers by influx of 4 mM Mg²⁺. **f** DA1 chromatin fibers compact dynamically by influx of 4 mM Mg²⁺ at 5 s as reported by a rapid increase in FRET. Displayed: Overlay of indicated number of traces from single fibers. Only traces exhibiting a FRET change were included in the analysis (65%). **g** DA1 chromatin decompacts rapidly upon removal of Mg²⁺ by injection of low-salt buffer/EDTA. Only traces exhibiting a FRET change were included in the analysis (74%)

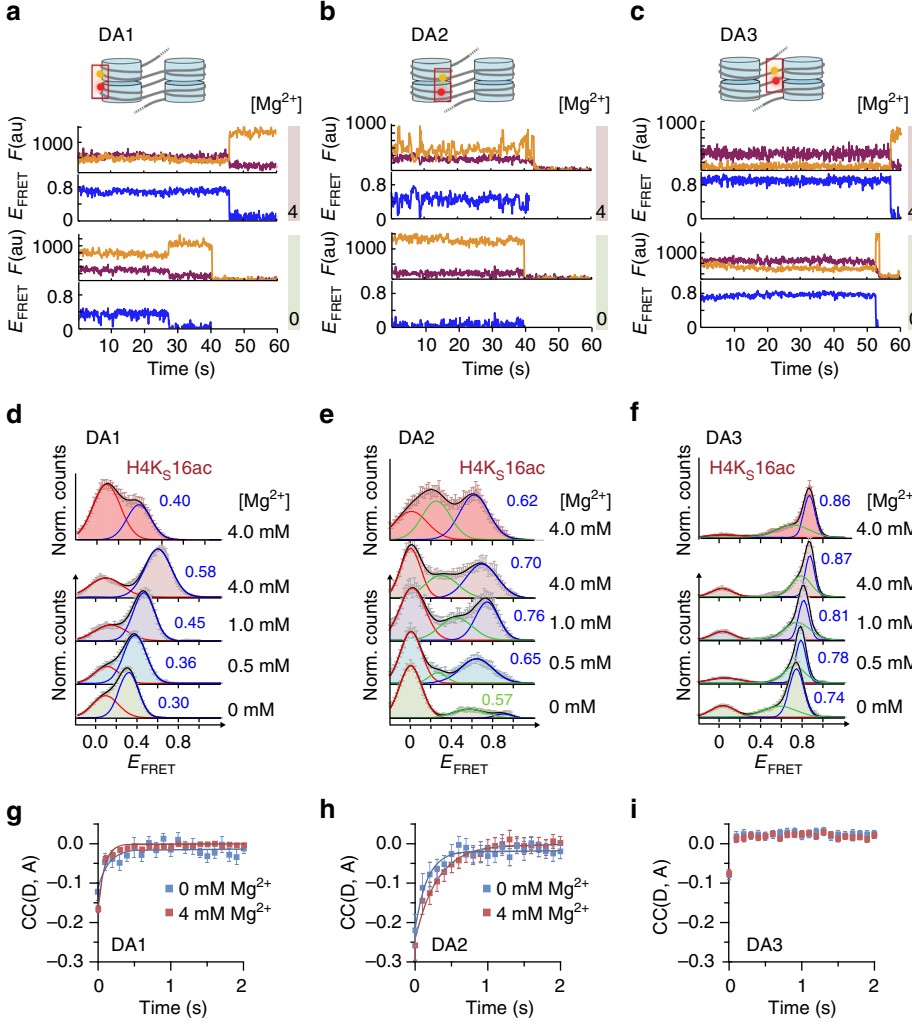

**Fig. 2** Multi-perspective smTIRF–FRET reveals dynamic chromatin compaction. **a** Single-molecule traces (donor: orange, acceptor: red, FRET: blue) for DA1 at 0 mM $Mg^{2+}$ (bottom), 4 mM $Mg^{2+}$ (top) until either donor or acceptor dye photobleaching. For analysis methods, see Supplementary Note, step 1 **b** FRET traces for DA2, same conditions as in **a**. **c** FRET traces for DA3, same conditions as in **a**. **d** FRET populations observed for DA1 at the indicated $Mg^{2+}$ concentrations, as well as in the presence of H4K$_S$16ac. **e** FRET populations for DA2, same conditions as in **d**. **f** FRET populations for DA3, same conditions as in **d**. **d**–**f** Error bars: s.e.m. For the number of traces, parameters of the Gaussian fits, see Supplementary Table 5. **g** Donor–acceptor channel cross-correlation analysis of DA1. Fits, 0 mM $Mg^{2+}$: cross-correlation relaxation time $t_R = 140 \pm 101$ ms ($n = 76$); 4 mM $Mg^{2+}$: $t_R = 73 \pm 13$ ms ($n = 229$). **h** Donor–acceptor channel cross-correlation analysis of DA2. Fits, 0 mM $Mg^{2+}$: $t_R = 169 \pm 79$ ms ($n = 61$); 4 mM $Mg^{2+}$: $t_R = 312 \pm 108$ ms ($n = 52$). **i** Donor–acceptor channel cross-correlation analysis of DA3. **g**–**i** Error bars: s.e.m. For the number of traces, see Supplementary Table 5. Fit uncertainties correspond to 95% confidence intervals of a global fit of the indicated number of traces. For the percentage of dynamic traces, see Supplementary Table 6

in a multivalent fashion. HP1α is a key architectural protein and is involved in establishing a compact chromatin state, thereby contributing to gene silencing[22–28]. Importantly, it has been revealed that HP1α is highly dynamic, with residence times on chromatin from milliseconds to seconds[23,27,29,30]. Thus, it is not clear how HP1 proteins induce chromatin compaction. Moreover, no detailed information is available about the internal structure of such compact states. The lack of precise information on chromatin dynamics in general, and of chromatin-effector complexes in particular, is mainly due to experimental constraints arising from the megadalton scale, molecular complexity, and structural heterogeneity of chromatin. Knowledge of the timescale of chromatin structural rearrangements, modulated by histone PTMs or by chromatin effectors[21,25,31,32], is however central for understanding the role of chromatin in gene regulation.

In this study, we combine two single-molecule Förster resonance energy transfer (smFRET)[33] methods, covering detection timescales from microseconds up to seconds, to directly map local

chromatin structural states and measure their interconversion dynamics. We fluorescently label chromatin fibers at three distinct sets of internal positions yielding structural information from several vantage points. Using two fluorescent dye pairs with different distance sensitivities (i.e., Förster Radii, $R_0$) allows us to measure a wide range of inter-dye distances ($R_{DA}$) with sub-nm precision. Employing this multipronged approach combined with dynamic structural biology methods (building on our FRET positioning and screening toolkit, FPS)[34], we identify distinct structural states in chromatin fibers and determine their exchange kinetics. We reveal that nucleosomes engage in stacking interactions, which rapidly interchange on the micro- to millisecond timescale. HP1α binding to modified chromatin fibers results in a compact but dynamic chromatin state, as HP1α transiently stabilizes stacked nucleosomes. Together, our study establishes a dynamic-register model of local chromatin fiber motions regulated by effector proteins.

## Results

**Reconstitution of site-specifically labeled chromatin fibers.** A key prerequisite for our smFRET studies is the introduction of a single dye pair with base-pair precision into chromatin fibers. We thus developed a method to assemble chromatin DNA constructs containing 12 copies of the "601" nucleosome positioning sequence[35] separated by 30 bp linker DNA. We used preparative ligations of two recombinant and three synthetic fragments, the latter of which carried the fluorescent labels (Fig. 1b, Supplementary Figs. 1–3, and Supplementary Tables 1–4). A convergent DNA assembly procedure with intermediate purification steps ensured the efficient and accurate incorporation of exactly one donor and one acceptor dye into chromatin DNA at defined positions. Guided by structural modeling[6,7,17], we decided on three dye configurations (Donor–Acceptor position 1, DA1), DA2 and DA3 (Fig. 1b), employing Alexa Fluor 568 (Alexa568) as FRET donor and Alexa Fluor 647 (Alexa647) as FRET acceptor. This pair has the advantage of a large Förster radius $R_0 = 82$ Å, enabling measurement of large inter-dye distances (up to 150 Å). Each dye pair was positioned in the center of the 12-mer nucleosome array (N1–N12) to probe distinct contacts and motions (Fig. 1a, b). DA1 senses stacking between nucleosomes N5 and N7 at a position close to the H2A–H2B four-helix bundle contacts[17]. DA2 measures inter-nucleosome interactions closer to the dyad (N5–N7). DA3 reports on dynamic modes within the linker DNA flanking the central nucleosome (N6). Chromatin fibers were reconstituted on double-labeled DNA templates (either DA1, DA2, or DA3) using recombinant human histone octamers (Supplementary Fig. 4). Ensemble measurements confirmed that all three dye configurations in chromatin resulted in increasing FRET as a function of magnesium-induced compaction, compatible with a two-start fiber model[6,7] (Supplementary Fig. 5a–l). Chromatin fibers labeled on nucleosome positions N5 and N6 (nearest-neighbor in sequence), which only make contact in a one-start fiber configuration, did not demonstrate measurable FRET. This finding, together with structural modeling, ruled out that solenoid or one-start fiber structures contribute to the measured FRET signal (Supplementary Fig. 5m–o).

**smFRET reveals structural heterogeneity in chromatin fibers.** We proceeded to investigate the conformational and dynamic properties of the assembled chromatin fibers using single-molecule imaging. In a first set of experiments, we applied single-molecule total internal reflection fluorescence (smTIRF) microscopy with a time resolution of 100 ms, to investigate chromatin structure and dynamics on the millisecond to seconds timescale (Fig. 1c). We immobilized DA1-labeled chromatin fibers in flow channels and measured their donor and acceptor fluorescence emission (Fig. 1d). Traces were selected according to a predefined set of selection criteria, e.g., the presence of a donor and an acceptor dye, and a minimal trace length in time (Supplementary Fig. 6g). We then generated time traces of FRET efficiency ($E_{FRET}$) (Supplementary Note, step 1). Chromatin compaction induced by rapid injection of bivalent cations (4 mM $Mg^{2+}$) resulted in a fast (<0.5 s) increase in $E_{FRET}$ (Fig. 1e, f). Conversely, rapid removal of $Mg^{2+}$ ions induced chromatin decompaction on a similarly rapid timescale (Fig. 1g). We can thus directly observe real-time conformational changes in single chromatin fibers. Moreover, these experiments reveal that the formation of chromatin higher-order structure occurs on the millisecond timescale.

Next, we systematically explored chromatin conformational changes as a function of bivalent cation concentration (0, 0.5, 1.0, and 4.0 mM $Mg^{2+}$) from our three structural vantage points (Fig. 2). We recorded time traces of FRET efficiency for DA1 (Fig. 2a), DA2 (Fig. 2b), and DA3 (Fig. 2c), which demonstrated an increase in $E_{FRET}$ with $Mg^{2+}$ for all positions, albeit to different extents. For DA1, $E_{FRET}$ histograms revealed a broad FRET distribution, which could be described with two Gaussians centered at low (<0.1) and intermediate (0.3–0.6) FRET efficiency values (Fig. 2d). In contrast, DA3 and DA2 showed a more complex pattern with one population at low $E_{FRET}$ and at least two populations associated with intermediate-to-high FRET efficiency (Fig. 2e, f). With increasing $Mg^{2+}$ concentration, for all arrays (DA1–3) the populations with $E_{FRET} > 0.1$ gradually shifted to higher FRET efficiency values, indicating an induction of nucleosome stacking.

As a confirmation that we indeed measured nucleosome stacking, we investigated the effect of acetylation of H4 at K16, which has been shown to abolish a key contact between the H4 tail and the H2A acidic patch of the neighboring nucleosome[31]. We thus synthesized a close chemical analog of this modification, H4K$_S$16ac (Supplementary Fig. 7). Inclusion of H4K$_S$16ac resulted in a significant reduction in internucleosomal stacking contacts observed by DA1 (Fig. 2d). A reduction in nucleosomal contacts was also registered by DA2 (Fig. 2e), whereas DA3 did not demonstrate a measurable change compared to the unmodified fiber (Fig. 2f). Thus, H4K16 acetylation results in a loss of defined and stable nucleosome stacking by disrupting a key internucleosomal interaction, while keeping the overall fiber geometry intact.

Considering unmodified chromatin fibers, we further resolved anti-correlated fluctuations in the time traces of donor and acceptor fluorescence emission (Fig. 2a–c), in particular for DA2, indicating structural dynamics. Cross-correlation analysis of donor and acceptor fluorescence fluctuations [CC(D,A)] revealed structural motions for DA2 positions (relaxation time $t_R = 0.2–0.3$ s, Fig. 2h), fast dynamics at the detection limit for DA1 ($t_R \sim 0.1$ s, Fig. 2g) and quasistatic behavior for DA3 (Fig. 2i). Together, the data from DA1–3 point toward complex multiscale dynamics featuring multiple FRET species in rapid exchange, which are not clearly resolvable with smTIRF.

**Chromatin fibers exist in two structural registers.** We thus employed a second approach, smFRET with confocal multiparameter fluorescence detection (MFD)[36], to study freely diffusing single chromatin fibers (Fig. 3a). This method extends the accessible dynamic timescale to the sub-microsecond range and resolves structural states with sub-nm accuracy[34]. For a set of excitation lasers (485 and 635 nm), our experimental setup allowed the application of pulsed interleaved excitation (PIE)[37] to filter out detections arising from donor-only molecules. To analyze MFD data, each photon burst (i.e., a single-molecule detection) is plotted in a 2D histogram as a function of two FRET indicators: the intensity-derived $E_{FRET}$ and the average (fluorescence-weighted) donor lifetime $\langle \tau_{D(A)} \rangle_F$ (Fig. 3b, c). As an example, molecules with two conformational states A and D, which remain static during their passage through the confocal volume are located as two populations on a static FRET line (dark red line, Fig. 3b). In contrast, molecules undergoing structural exchange dynamics with a characteristic relaxation time $t_R$ between the limiting structural states A and D are detected by a broadened intermediate peak, reminiscent of NMR signals in the intermediate exchange regime (Fig. 3c). Moreover, these dynamically broadened populations are located on a dynamic FRET line (blue line, Fig. 3c), which connects the limiting FRET species involved in the fast exchange (intersection of blue and red line in Fig. 3c)[38].

We performed MFD measurements for chromatin fibers carrying FRET dye pairs in configurations DA1–3 (exciting at

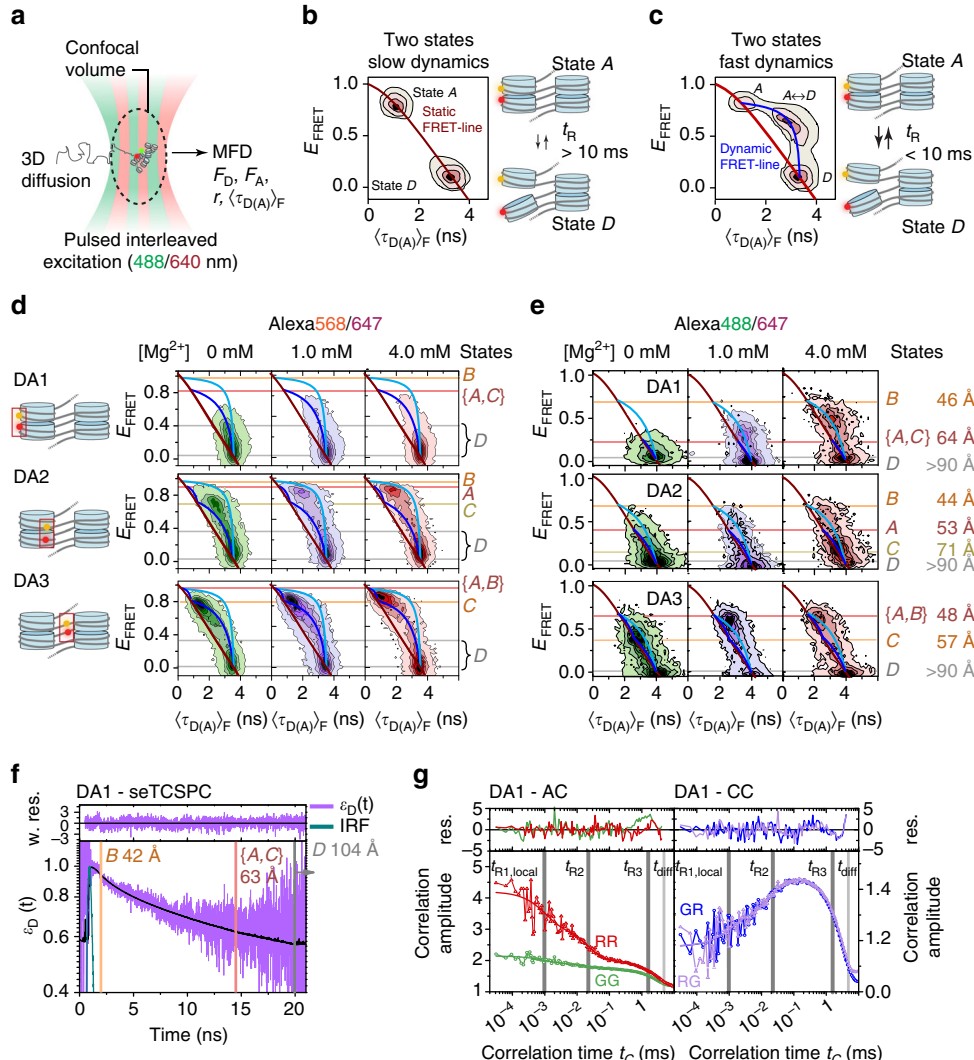

**Fig. 3** Multiscale chromatin dynamics in two registers revealed by MFD. **a** Scheme of PIE-MFD: Species-averaged donor and acceptor emission intensities ($F_D$, $F_A$), intensity-averaged donor lifetime $\langle \tau_{D(A)} \rangle_F$ and anisotropy ($r$) are simultaneously measured for each molecule diffusing through the confocal volume. **b** Principle of MFD analysis: If dynamics between two states A and D are slow (relaxation time $t_R \gg 10$ ms), distinct structural states are resolved by $E_{FRET}$ and $\langle \tau_{D(A)} \rangle_F$, falling on the static FRET line (red). **c** Fast dynamics ($t_R < 10$ ms) result in an intermediate peak (labeled A ↔ D) on a dynamic FRET line (blue). Peak shape analysis reveals $t_R$ (Fig. 5). **d** 2D-MFD histograms for chromatin fibers DA1–3 (Alexa568/647) at indicated $Mg^{2+}$ concentrations. These histograms contain contributions from donor-only labeled chromatin fibers. Red line: static FRET line. Dark and bright blue lines: Two dynamic FRET lines for the two tetranucleosome registers 1 and 2, indicating dynamic exchange with $t_R < 10$ ms (for parameters of all FRET lines, see Supplementary Note, step 2). Red, orange, yellow, and gray lines: FRET species A–D (see also Fig. 4a). **e** 2D-MFD histograms for chromatin fibers DA1–3 labeled with Alexa488/ 647 at indicated $Mg^{2+}$ concentrations. Red line: static FRET line; dark and bright blue lines: dynamic FRET lines. **f** Subensemble fluorescence lifetime analysis for DA1-labeled fibers (Alexa488/647) at 1 mM $MgCl_2$ and $E_{FRET} > 0.065$. The FRET-induced donor decay $\varepsilon_D(t)$ was fitted with contributions from FRET species {A, C}, B and D (for details and fit parameters of eqs. 3.1–6, see Supplementary Note, step 3), corresponding to the indicated inter-dye distances. IRF: instrument response function. **g** Auto- (left panel) and cross (right panel)-correlation functions of the donor (G) and acceptor (R) emission channels for the same subensemble as in **f**. Global analysis of FCS curves reveals FRET dynamics with two global structural relaxation times ($t_{R2} = 27$ μs (27%); $t_{R3} = 3.1$ ms (56 %)), a term describing local fluctuations ($t_{R1,local} = 2.6$ μs (17%)) and an apparent diffusion time for all curves ($t_{diff} = 4.96$ ms) (for details and fit parameters of Eq. 4.1, see Supplementary Note, step 4)

530 nm, which precluded PIE), which revealed a complex population distribution involved in dynamic exchange (Fig. 3d) not observed in free DNA or donor-only labeled chromatin fibers (Supplementary Fig. 8c, d). Due to the absence of PIE in those measurements, donor-only labeled chromatin fibers ($E_{FRET} = 0$) contributed also to the total observed signal. An iterative 11-step workflow (Supplementary Fig. 9) allowed us to resolve distinct structural states by their characteristic FRET efficiencies and dynamics. Based on this analysis, the data could only consistently be described by two dynamic FRET lines (dark and bright blue

lines, Fig. 3d), indicating two coexisting subpopulations of dynamic chromatin fibers, which are distinct within the observation time of ~10 ms.

From the intersections of the dynamic with the static FRET lines, we identified four limiting FRET species involved in the exchange: A, B, C, and D, indicated by the horizontal lines in Fig. 3d. Braces (e.g., {A, C}) indicate conformational states sharing indistinguishable FRET efficiencies. Importantly, a complementary analysis procedure within our workflow (Supplementary Fig. 9), subensemble fluorescence lifetime analysis,

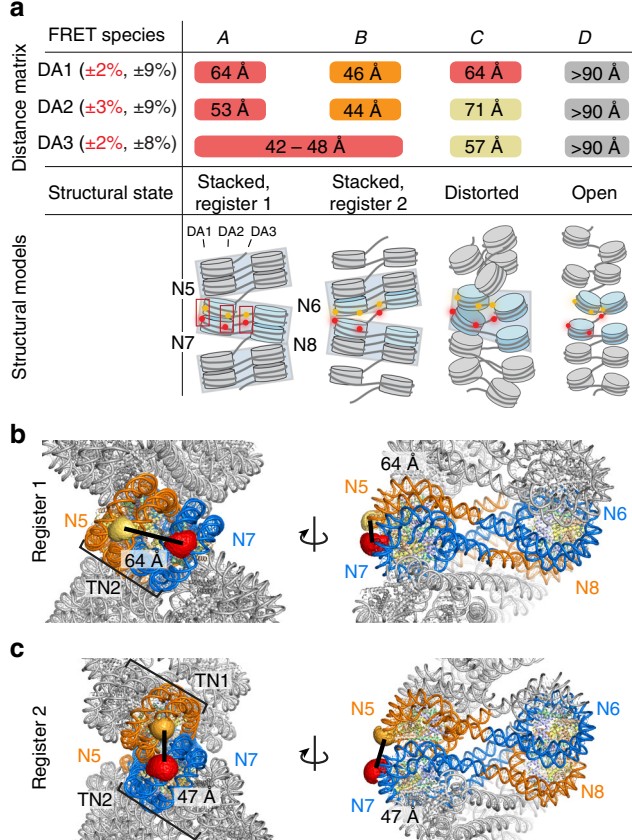

**Fig. 4** Chromatin fibers exist in two rapidly interchanging tetranucleosome stacking registers. **a** Matrix of the inter-dye distances $R_{DA}$ for DA1, DA2, and DA3 obtained from dynPDA. Species that cannot be discriminated with a given FRET pair are labeled with the same color and/or a continuous box. Percentages given in brackets: uncertainties in the observed distances. Red: precision ($\Delta R_{DA}(R_{DA})$), relevant for relative $R_{DA}$, calculated as s.d. between three PDA analyses of data sets comprising a fraction (70%) of all measured data (subsampling). Black: Absolute uncertainty, mainly determined by the uncertainty in $R_0$ (Supplementary Note, step 9 and Supplementary Table 7). The combined average inter-dye distances $R_{DA}$ over DA1–3 allow us to map each FRET species to a class of corresponding structural states of chromatin (Supplementary Figs. 12 and 13, Supplementary Table 8, and Supplementary Note, steps 9 and 10). The registers of tetranucleosome units are indicated by light gray boxes. **b** Structural model of a chromatin array, consisting of a stack of three tetranucleosomes (register 1) with DA1-positioned dyes in the central tetranucleosome, based on ref. 7. The inter-dye distance was evaluated using simulated dye accessible contact volumes (ACV)[34]. **c** Molecular structure of a chromatin array, consisting of a stack of two tetranucleosomes, flanked by two unstacked nucleosomes at each side (register 2) with DA1-positioned dyes on the two central tetranucleosomes and inter-dye distance from ACV calculations. Molecular models for DA2 and DA3 are reported in Supplementary Figs. 12 and 13

corroborated the FRET species for each labeling pair DA1–3. Similarly, model-free fluorescence correlation analysis from DA1–3 revealed conformational dynamics with at least three relaxation times, thus involving at least four kinetic species (A–D). Finally, the FRET line parameters were determined in independent experiments[38] (see Supplementary Note, step 2).

In summary, for all vantage points DA1–3 our analysis revealed compact chromatin fibers ($E_{FRET}>0.8$) in rapid exchange with extended structures (Fig. 3d). At least two independent dynamic transitions were consistently resolved, as indicated by

the two dynamic FRET lines, revealing distinct limiting FRET species with high $E_{FRET}$ (compact species, A–C) and with very low $E_{FRET}$ (open species, D), respectively. The existence of two dynamic transitions, as indicated by the two FRET lines, directly revealed two populations of chromatin fibers. Each population shows unique internal exchange dynamics but no interchange between the populations is observed during the ~10 ms observation time. Chromatin fibers are thus structurally and dynamically heterogeneous at the local level.

**Revealing structural states in dynamic chromatin fibers**. To delineate the fiber architectures corresponding to these populations, we performed MFD experiments using Alexa Fluor 488 as a FRET donor ($R_0 = 52$ Å). This FRET donor substantially improved the spatial resolution at shorter distances (Supplementary Fig. 8b). Importantly, excitation at 485 nm enabled us to employ PIE. We thus could exclude donor-only labeled chromatin fibers. In agreement with the previous MFD measurements (Fig. 3d), FRET distributions were also located on two dynamic FRET lines (Fig. 3e). Due to the altered distance sensitivity of the Alexa488/647 FRET pair, compact states (A, B, and C) were now better resolved. As a result, in these experiments the dynamic FRET lines fell closer to the static FRET lines (while remaining well defined), as compared to measurements with Alexa568/647. Together, these measurements with two different labeling schemes confirm the existence of four structural states in two distinct fiber populations interchanging with fast internal dynamics.

Subensemble fluorescence lifetime analysis provides an alternative method to directly resolve the individual FRET efficiencies (and thus $R_{DA}$ values) within a dynamic ensemble. In effect, it provides a nanosecond snapshot of the coexisting FRET species, independent of their exchange dynamics. We thus averaged photon bursts from DA1 (selecting only bursts with $E_{FRET}>0.065$) and computed a FRET-induced fluorescence decay of the donor $\varepsilon_D(t)$ (Fig. 3f and Supplementary Note, step 3)[39]. The nonlinear decay of $\varepsilon_D(t)$ on a log scale directly demonstrated the existence of at least three FRET species. We employed a global analysis to resolve the inter-dye distances characteristic for the three corresponding FRET species {A, C}, B and D (Fig. 3f and Supplementary Fig. 10), closely matching the limiting FRET states observed in 2D-MFD histograms (Fig. 3d, e).

Fluorescence correlation analysis enables a direct and model-free assessment of molecular dynamics. We thus analyzed the autocorrelation functions for the donor and acceptor channels, as well as the cross-correlation between donor and acceptor fluorescence channels (Fig. 3g, and Supplementary Fig. 11, and Supplementary Note, step 4). For DA1, this analysis directly confirmed the existence of structural dynamics between the FRET species {A, C}, B and D, revealing two slow kinetic exchange processes with relaxation time constants $t_R$ of 27 µs and 3.1 ms. However, solely based on this analysis, the relaxation times could not be attributed to individual conformational dynamics.

**Resolving conformational dynamics in chromatin fibers**. An integrated approach is required to characterize the two dynamic populations in chromatin fibers, and to resolve their underlying structural states. We thus proceeded along our workflow for dynamic structural biology (Supplementary Fig. 9 and Supplementary Note): Using the combined information from TIRF measurements, MFD histograms, subensemble lifetime analysis, and fluorescence correlation analysis for DA1–3, we were able to analyze the experimental data with dynamic photon distribution analysis (dynPDA)[38] (Supplementary Note, steps 6–8). This is an approach comparable to the analysis of NMR relaxation

dispersion experiments, resolving subpopulations and their exchange dynamics. While dynPDA is an inherently iterative method, for clarity we first address structural considerations followed by a discussion of the observed dynamics.

Our dynamic–structural biology approach revealed high-precision inter-dye distances (displayed as a distance matrix in Fig. 4a) for species (A–D) with respect to the three vantage points of the samples DA1–3 (Fig. 4a). Using the recovered inter-dye distance sets as constraints, we assigned molecular structures to species (A–D), based on available high-resolution structural data[6,7] and coarse-grained simulations[40] (Fig. 4b, c, Supplementary Figs. 12 and 13, and Supplementary Note, steps 9 and 10). Distance constraints from DA1 and DA2 showed that FRET species A and B correspond to conformational states with defined tetranucleosome units in two different interaction registers relative to the FRET labels. Register 1 (A) positions the label pairs in the same tetranucleosome unit (Fig. 4a, b). This chromatin fiber conformation is consistent with the reported cryo-EM structure of a 12-mer chromatin fiber[7]. Register 2 (B) positions the FRET labels across two neighboring tetranucleosome units, indicating a fiber structure that exhibits altered nucleosome interactions (Fig. 4a, c). Species (C) corresponds to a distorted (twisted) tetranucleosome state within register 1. Finally, species (D) corresponds to an ensemble of open chromatin fiber conformations.

From the DA1 vantage point, the two compact species (A) and (C) shared a single inter-dye distance, resulting in the apparent FRET species {A, C}. This can be rationalized as the DA1 dye pair is close to a key internucleosomal interaction, mediated via the H2A–H2B four-helix bundle[6,7]. This interaction restricts local internucleosomal motions. DA2, in contrast, detected the distorted tetranucleosome state (C), which for this vantage point exhibits an increased inter-dye distance. Hence, stacked nucleosomes exhibit more structural flexibility close to the dyad. Finally, all three dye pairs DA1–3 reported on the species (D), accounting for open chromatin devoid of local internucleosomal interactions.

**A dynamic register model for chromatin dynamics**. To uncover fundamental motions within chromatin fibers, the kinetic connectivity of the chromatin structural states must be elucidated. We thus employed all the previously discussed information to formulate kinetic models, which were employed to fit the experimental FRET efficiency histograms by dynPDA (Fig. 5a–c, Supplementary Fig. 14, and Supplementary Note, steps 5–8). To find an appropriate kinetic model, we performed global fits over the $Mg^{2+}$ dependence for each data set DA1–3. We tested a set of 3- and 4-state kinetic models describing distinct kinetic connectivities between species (A–D) (Fig. 5d–f and Supplementary Fig. 15). In agreement with two dynamic populations detected in MFD plots, a successful and consistent fit for all label pairs was achieved with a kinetic model containing two branches: one branch connecting species (A, C) to (D), the second branch connecting species (B) to (D) (Fig. 5d–f and Supplementary Figs. 16–18). The revealed kinetic information provided insights into the dynamics of chromatin fibers: an analysis of DA1 (Fig. 5d) indicated that stacked nucleosome (A, register 1) exchange with open conformations (D) with a relaxation time $\tau_R$ = 3.7 ± 0.3 ms (uncertainties of relaxation times: s.d. between three PDA analyses of data sets comprising a fraction (70%) of all measured data (subsampling)). These motions are two orders of magnitude slower compared to fluctuations between tetranucleosomes (B–D, register 2, $\tau_R$ = 60 ± 10 μs). This is consistent with the significant free energy (around 13 kT) associated with nucleosome stacking[20]. DA2 provided further insight into intra-tetranucleosome dynamics (Fig. 5e), where structural distortions

(i.e., torsional fluctuations and partial nucleosome disengagement, species C) occur on a 0.5 ± 0.06 ms timescale, followed by a transition to D within 2.6 ± 0.5 ms. DA3, finally, reported on linker DNA fluctuations (Fig. 5f). Here, we detected increased (C–D) transition rates, indicating a contribution from transient DNA unwrapping dynamics[12,13,41]. Analyzing the populations of species A–D for DA1–3 over the range of $Mg^{2+}$ concentrations revealed a coherent picture of the dynamic chromatin structure (Fig. 5g–i): Compact conformers in register 1 (A, C) were about twice as highly populated as register 2 contacts (B). Thus, register 1 with maximally three formed tetranucleosomes is energetically more favorable than register 2 that can only encompass two stacked tetranucleosome units. Compact conformers were increasingly more populated at higher bivalent ion concentrations, but remained in rapid exchange with open and compact chromatin. Finally, between 20 and 40% of all observed chromatin fibers did not show any measurable dynamics on the MFD timescale (observed for all species (A–D), see Supplementary Figs. 16–18). This indicates the presence of chromatin structures separated by significant barriers from the rapidly exchanging structural ensemble (locked states), consistent with the observation of slow dynamics in TIRF-FRET measurements.

Together, our smFRET measurements revealed intriguing multiscale chromatin dynamics across five orders of magnitude in time. We propose a unified model (the dynamic-register model) to describe higher-order chromatin structure and its local dynamics (Fig. 6 and Supplementary Note, step 11). In a chromatin fiber a nucleosome can, at any time, engage in tetranucleosome contacts with only one of its two neighbors within the two-start helix. On a short range, this results in at least two interchanging interaction registers. The exchange pathway between registers 1 and 2 always leads through local fiber unfolding and subsequent reformation of the (altered) tetranucleosome contacts.

A chromatin fiber has more conformational degrees of freedom than those directly probed by FRET in this study. Thus, we use structural and dynamic features to subdivide the observable FRET species A–D further into an ensemble of conformational states (indicated by the numerical index in Fig. 6). Fluctuations observable in smTIRF–FRET (and quasistatic molecules in MFD) indicate the existence of nucleosome interactions stable for a few hundreds of milliseconds (locked states $A_1$, $B_1$) as well as dynamic species (unlocked states $A_2$, $B_2$). In register 1, we observed rearrangements of the nucleosome interface allowing tetranucleosomes to open on a millisecond timescale (to $A_3$, C, and the ensemble of open states $D_n$). In contrast, neighboring tetranucleosomes in register 2 are only loosely associated, resulting in sub-millisecond interaction dynamics governed by shallow energy barriers ($B_2$ to $D_1$ and $D_n$). Importantly, this dynamic ensemble of higher-order structures (or supertertiary structure[42]) with multiple conformational states and dynamic transitions is a fundamental property of chromatin fibers. Elementary states are observed both in extended and compact fibers, but are populated to different extents. Our analysis thus suggests that these elementary states and their transitions govern the biochemical accessibility, regulation, and biological function of chromatin.

**HP1α induces a dynamically compacted chromatin structure**. Having established this dynamic model of chromatin, we asked how HP1α affects the internal structure and dynamics of chromatin fibers. Previous studies indicated that HP1α can compact chromatin[25,43] and that it can cross-bridge H3K9me3-modified nucleosomes[28]. However, no information was available about the internal structure of HP1α-complexed chromatin. Single-

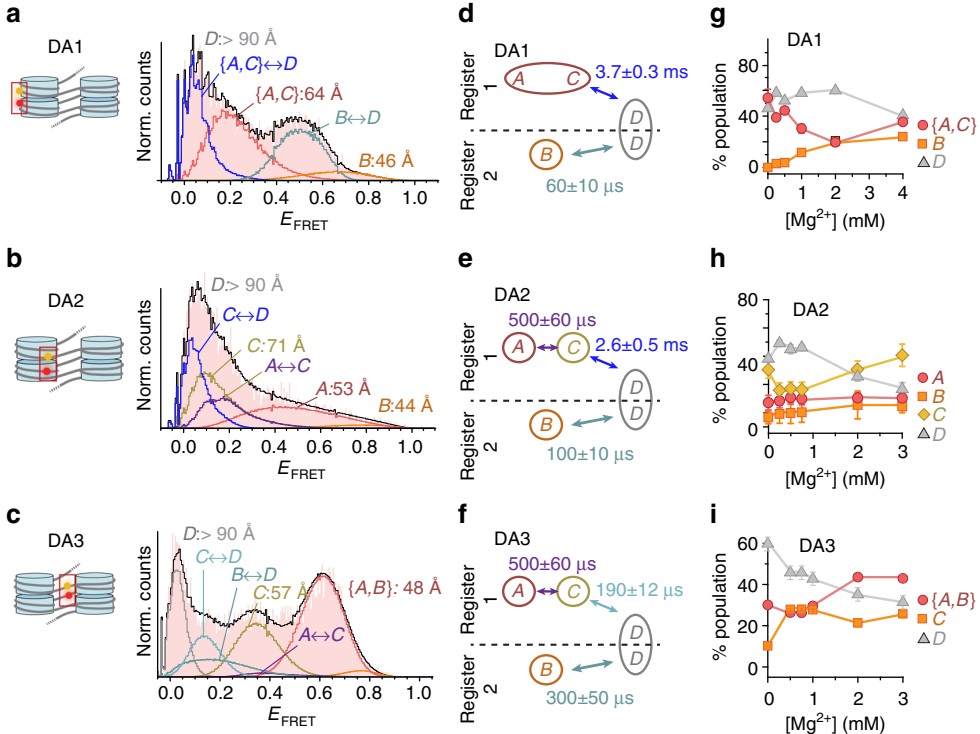

**Fig. 5** Chromatin exhibits multiscale dynamics. **a–c** dynPDA analysis of MFD data (for a detailed description, see Supplementary Note, steps 6–8). Red histogram: Experimental data, black line: PDA fit to the kinetic models corresponding to the indicated state connectivities (Fig. 5d–f). Gaussian distributions in orange hues or gray: Distributions corresponding to FRET states indicated in Fig. 4a: A (red), B (orange), C (yellow), and D (gray). Blue hues: Distributions originating from dynamic exchange between FRET species: A ↔ C (violet), C ↔ D (dark blue), B ↔ D (gray blue). **a** dynPDA analysis of MFD data for DA1 (at 4 mM Mg$^{2+}$) using the kinetic connectivity outlined in Fig. 5d. **b** dynPDA analysis of MFD data for DA2 (at 3 mM Mg$^{2+}$) using the kinetic connectivity outlined in Fig. 5e. **c** dynPDA analysis of MFD data for DA2 (at 3 mM Mg$^{2+}$) using the kinetic connectivity outlined in Fig. 5f. **d–f** Kinetic connectivity maps for DA1–3 used for dynPDA, which describe the experimental data. Two dynamic equilibria (registers) are observed: Register 1 comprises species A, C, and D (as characterized by their inter-dye distance, $R_{DA}$), exchanging with the indicated relaxation times. Register 2 comprises species B and D in equilibrium. Register exchange within D is not permitted in the model on the investigated timescales, as indicated by the dashed line. The indicated time constants are given for 2 mM Mg$^{2+}$. For the individual rate constants, see Supplementary Figs. 16–18. Uncertainties: s.d. between three PDA analyses of data sets comprising a fraction (70%) of all measured data (subsampling). **g** Relative combined populations of observed species A–D for DA1 as a function of [Mg$^{2+}$] (for the individual contributions of static and dynamic molecules, see Supplementary Figs. 16–18). **h** Relative combined populations for species A–D for DA2. (**i**) Relative combined populations for species A–D for DA3. For the full PDA fits, see Supplementary Figs. 16–18. Error bars: s.d. between three dynPDA analyses of data sets comprising a fraction (70%) of all measured data (subsampling). In some cases, the error bars are smaller than the symbol size

molecule-binding studies revealed that HP1α interacts with chromatin on the 250 ms timescale[27], matching the time resolution of our FRET-TIRF approach. We thus reconstituted DA1 and DA2 chromatin fibers containing either unmethylated (H3K9me0) or chemically produced H3K9me3 (Supplementary Fig. 19a, b) and measured smFRET in the presence of 1 μM HP1α using TIRF microscopy (Fig. 7a, b). The presence of HP1α resulted in H3K9me3-dependent chromatin compaction as observed by an increase in $E_{FRET}$ from the vantage points DA1 and, in particular, from DA2 (Fig. 7c, d). The larger effect on DA2 indicates that HP1α stabilizes nucleosome stacking primarily toward the center of the chromatin fiber, where the FRET efficiency reaches the value ($E_{FRET} > 0.8$) of the limiting species A, B resolved by MFD measurements (Fig. 3d). This comparison directly shows that the HP1α-compacted state involves the same inter-nucleosome contacts as observed in the absence of HP1α.

HP1α is post-translationally modified in particular by phosphorylation of its N-terminal extension (NTE)[44]. Intriguingly, this modification not only stabilizes H3K9me3 binding[45–47] leads to HP1α oligomerization and phase separation behavior important for heterochromatin establishment[48,49]. We thus produced phosphorylated HP1α (pHP1α, Supplementary Fig. 19f–i). Phosphorylation indeed increased the compacting

effect by stabilization of nucleosome binding and by strengthening HP1α interactions beyond the dimer (Fig. 7c, d). Intriguingly, the analysis of FRET traces by cross-correlation analysis of donor and acceptor fluorescence revealed high-amplitude dynamic fluctuations with a sub-second relaxation time in the presence of HP1α (Fig. 7e, f). Thus, chromatin fibers compacted by HP1α do not adopt a stably closed conformation, but in contrary remain highly dynamic and exhibit structural fluctuations on the sub-second timescale.

Finally, we wondered how fast HP1α could compact chromatin fibers. We thus injected 1 μM HP1α into flow cells containing H3K9me3-modified chromatin fibers and monitoring FRET via the DA2 FRET pairs. The accumulated traces revealed an increase of compaction with a time constant of $1.1 \pm 0.4$ s (Fig. 7g, h, fit uncertainties correspond to 95% confidence intervals, global fit of $n = 86$ traces). Thus, HP1α needs to accumulate on chromatin to reach a critical density before compaction can take effect.

In summary, we find that HP1α transiently stabilizes interacting nucleosomes in chromatin fibers. This is most likely achieved by cross-bridging nucleosomes through H3K9me3 interactions[24,25,28] (Fig. 7i), a process which occurs on the hundreds of milliseconds timescale consistent with measured residence times for HP1α[27].

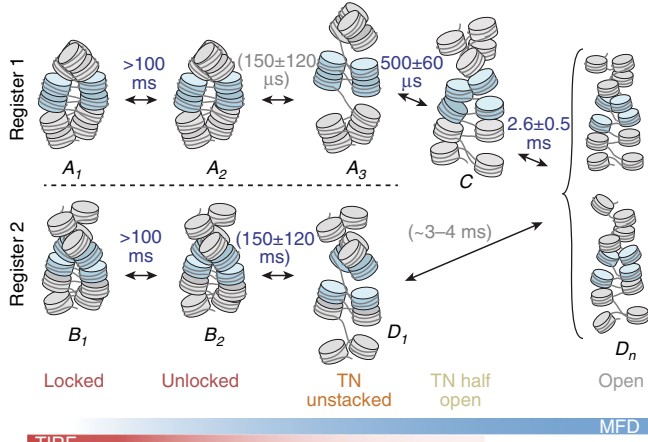

**Fig. 6** The dynamic register model of chromatin fiber dynamics (for details see text). The colored bars indicate the sensitivities of the two applied smFRET methods. The letters A, B, C, and D correspond to observed FRET species (Fig. 4a). Nucleosomes highlighted in blue are labeled and thus observed in the experiment. Numbered states correspond to different chromatin conformations, which exhibit the same FRET efficiency for DA1–3 but which can be kinetically differentiated. FRET species A includes conformational states {$A_1$, $A_2$, $A_3$} for which stacked nucleosomes are observed. FRET species B includes all states {$B_1$, $B_2$} corresponding to observation across two neighboring tetranucleosome units. FRET species D (low-FRET states) includes locally unstacked nucleosomes ($D_1$) and the ensemble of open fibers ($D_n$). Gray relaxation time constants are indirectly inferred; blue relaxation times are directly observed. The error ranges represent s.d. between observations of the same dynamic process with different FRET label pairs (for $B_2 \leftrightarrow D_1$), or directly from PDA subsampling (Fig. 5)

## Discussion

The structural dynamics of chromatin dictate biochemical access to the DNA and thus directly impinge on dynamic regulatory processes, such as TF binding, transcription, or DNA repair. A detailed knowledge of the structural states and exchange timescales within chromatin is therefore of critical importance. Previous experiments indicated that chromatin is highly dynamic[17–21], but stopped short of a detailed structural and kinetic exploration of unconstrained chromatin fibers.

Here, we employed two distinct smFRET approaches with access to complementary experimental timescales to reveal the structural and dynamic landscape of chromatin fibers. Based on our results, we formulated a dynamic-register model (Fig. 6) describing the fundamental dynamic modes governing biochemical access to compact chromatin. Our data are in agreement with the tetranucleosome as a fundamental unit of chromatin fibers[21]. We however discover a distinct set of motions within and between tetranucleosome units that introduce dynamic heterogeneity into chromatin structure. Individual tetranucleosomes can spontaneously open on the millisecond timescale. In contrast, interactions between neighboring tetranucleosomes fluctuate in the microsecond time regime. Neighboring tetranucleosomes can exchange their interaction register on the hundred millisecond timescale, by concerted unfolding, followed by refolding in the alternative register.

The existence of such a fundamental dynamic landscape of chromatin is analogous to the situation in proteins, where intrinsic motions govern function[50,51]. In chromatin, fiber dynamics are coupled to processes such as the target search of TFs, e.g., through sliding and hopping[52]. As these interaction modes require direct access to the DNA, local chromatin dynamics control the fundamental timescale of DNA sampling

and thereby set a speed limit for TF-binding kinetics. Intriguingly, direct observations of TF chromatin sampling in vivo reveal that these interactions occur on similar timescales as the local chromatin dynamics revealed in this work[53,54]. Finally, dynamic coupling mechanisms are not limited to TFs, but extend to other processes such as chromatin remodeling[55] or gene transcription itself[56].

Our measurements revealed that individual nucleosomes engage in short-lived (milliseconds to hundreds of milliseconds) stacking interactions with their neighbors, forming tetranucleosome units. Tetranucleosome contacts hinder access to linker DNA[6] and occlude the nucleosome acidic patch, the major interaction site for many chromatin effectors[57–59]. In agreement, structural[6,7] and force spectroscopy studies reported tetranucleosomes as basic organizational units of chromatin[21]. The observation of both a population of short-lived (milliseconds) as well as long-lived tetranucleosome states (locked states with lifetimes of hundreds of milliseconds) demonstrates that several inter-nucleosome interactions have to be released to allow rapid local fiber dynamics. One intriguing possibility is that long-lived (locked) states arise due to stabilizing long-range inter-nucleosomal interactions outside the tetranucleosome unit, which provide additional stability to chromatin fiber structure.

Importantly, we found that tetranucleosome contacts alternate between different registers on the 100-ms timescale. The interchange between registers requires cooperative motions between neighboring tetranucleosome units, at least over the range of four to eight nucleosomes. It is thus conceivable that structural disturbances in the fiber, e.g., by a bound TF, have long range effects on neighboring genomic loci by a modulation of DNA site exposure dynamics. Indeed, cooperative and collaborative effects between TF-binding sites have been observed over distances significantly exceeding a single nucleosome[60], pointing toward a role of long-range chromatin organization.

Several genome-wide studies have determined the existence and prevalence of tetranucleosome contacts in vivo, employing analysis of nucleosome contacts by electron microscopy[61], Micro-C[4] or in situ radical fragmentation of chromatin[5]. Long stretches of ordered chromatin structure are however not readily observed in interphase nuclei[62]. Our findings regarding the rapid dynamics and heterogeneity provide a rationale of this absence of order over large spatial and temporal scales. Rather, internucleosomal contacts are in constant exchange, forming local transient structures that are permissive for chromatin effectors.

The inherent flexibility and structural adaptability gives chromatin an inhomogeneous dynamic secondary structure with conformational fluctuations ranging over several orders of magnitude in time and space. This makes chromatin an ideal hub for interactions with diverse partners, including architectural such as H1, as well as a large range of chromatin effectors. Our developed methods for dynamic structural biology of chromatin enabled us to systematically determine local effects on such dynamic interactions.

Here we explored how HP1α, a key heterochromatin component, affects chromatin fibers depending on the presence of H3K9me3. We found that HP1α transiently stabilizes internucleosome contacts, most probably through multivalent engagement of two PTMs on different nucleosomes[27]. This results in an increased population of compact states, reducing local chromatin accessibility. In agreement, the presence of HP1α in vivo is correlated with increased tetranucleosome contacts[5].

Strikingly, HP1α-compacted chromatin fibers remained highly dynamic (Fig. 7i). First, HP1α interacts with DNA in addition to H3K9me3[28,46], which might directly modulate local chromatin motions. Second, HP1α has a stronger compacting effect around the nucleosome dyad. This suggests that the protein has a

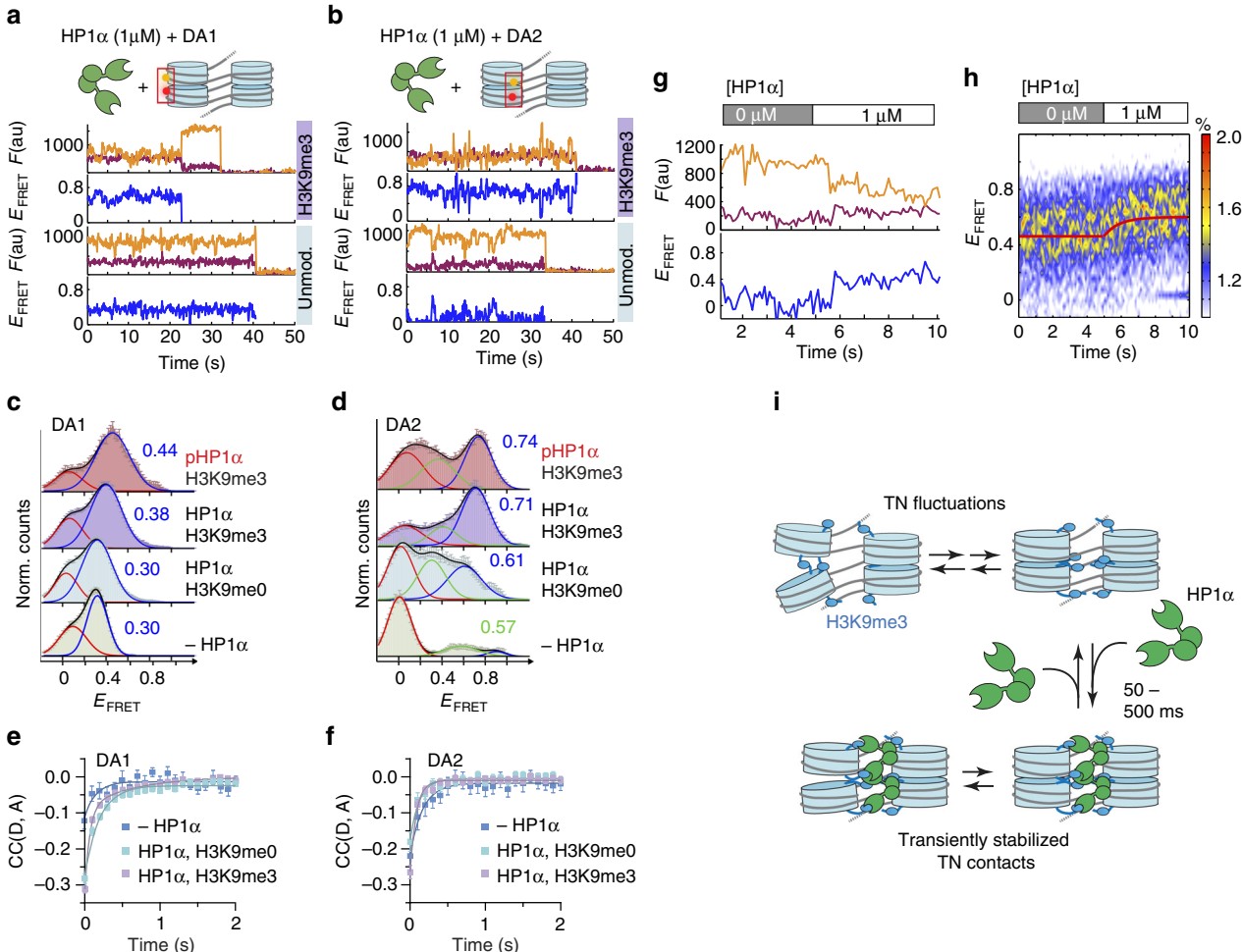

**Fig. 7** HP1α binding results in dynamically compacted chromatin. **a** FRET traces for DA1, containing no modification or H3K9me3 in the presence of 1 μM HP1α and the absence of Mg²⁺. **b** FRET trace for DA2, containing no modification or H3K9me3 in the presence of 1 μM HP1α. **c** FRET populations for DA1, showing H3K9me3-dependent compaction by HP1α and phosphorylated HP1α (pHP1α). **d** FRET populations for DA2, demonstrating close contacts induced by HP1α/pHP1α. **c**, **d** Error bars: s.e.m. For the number of traces, parameters of the Gaussian fits, see Supplementary Table 5. **e** Donor–acceptor channel cross-correlation analysis of DA1 in the presence of 1 μM HP1α. Fits, H3K9me0: $t_{R,1} = 200 \pm 25$ ms ($n = 530$), H3K9me3: $t_{R,1} = 64 \pm 13$ ms (72%), $t_{R,2} = 640 \pm 126$ ms (28%) ($n = 430$). **f** Donor–acceptor channel cross-correlation analysis of DA2 in the presence of 1 μM HP1α. Fits, H3K9me0: $t_R = 123 \pm 38$ ms ($n = 99$); H3K9me3: $t_{R,1} = 66 \pm 16$ ms (88%), $t_{R,2} = 930 \pm 543$ ms (12%) ($n = 106$). Fit uncertainties correspond to 95% confidence intervals of a global fit of the indicated number of traces. For the percentage of dynamic traces, see Supplementary Table 6. **e**, **f** Error bars: s.e.m. For the number of traces, see Supplementary Table 5. **g** Stochastic compaction of chromatin induced by injection of HP1α at 5 s. **h** 2D histogram of multiple injections. Only traces exhibiting a FRET change were included in the analysis (42%). The fit yields a time constant of $1.1 \pm 0.4$ s (fit uncertainties correspond to 95% confidence intervals, global fit of $n = 86$ traces). **i** Model of transient stabilization of tetranucleosomes, which still retain some internal flexibility, by HP1α

tendency to bind at central as opposed to peripheral sites within a chromatin fiber. Third, individual HP1α molecules do not remain stably bound to fibers, but exhibit rapid exchange dynamics in vitro[27] and in vivo[23,29,30] on the hundreds of millisecond to seconds timescale. Rapid HP1α turnover will thus stochastically release the stabilization of local nucleosome stacking interactions allowing local exposure of internal sites.

Functionally, the dynamic HP1α-compacted state remains permissive for biochemical access to the fiber, albeit to a reduced degree. Moreover, we expect bound HP1α to impair tetra-nucleosome register exchange, as this requires transient opening of two neighboring tetranucleosomes. Together, these effects therefore contribute to repression of transcription in heterochromatin. Nevertheless, as all DNA sites and nucleosome surfaces are eventually exposed, effectors such as pioneer TFs[63] or even the transcription machinery can still invade the hetero-chromatin state. In agreement, heterochromatin regions generally are transcribed at low levels[64]. Moreover, local accessibility makes

rapid chromatin regulation possible as a function of cellular stimuli[65].

In summary, our single-molecule studies reveal dynamic het-erogeneity within chromatin fibers, where the intrinsic dynamics are determined by a complex energy landscape. Dynamic higher order or supertertiary structure is governed by interactions of tetranucleosomes that form the fundamental structural units and provide local cooperativity through register exchange dynamics. Chromatin effectors, such as HP1α, selectively modulate this energy landscape by stabilizing specific conformations from the rapidly exchanging ensemble, thereby enacting a biological out-put. Thus, the mutual interplay between chromatin dynamics and effector proteins controls downstream biological processes.

## Methods

**Plasmid generation, purification, and DNA fragment isolation.** Plasmids for chromatin DNA production (recP1, recP5) were generated in DH5α cells grown in 6 L 2xTY medium and isolated by alkaline lysis followed by preparative gel

 

filtration as follows: After 18–20 h culture, cells were harvested by centrifugation, resuspended and homogenized in 80 mL alkaline lysis solution I (50 mM glucose, 25 mM Tris, 10 mM EDTA, pH 8.0). Homogenate was diluted to 120 mL with the same solution. An aliquot of 240 mL alkaline lysis solution II (0.2 M NaOH, 1% SDS) was added and mixed. An aliquot of 240 mL alkaline lysis solution III (4 M KAc, 2 M Acetic acid) was added to neutralize the solution followed by mixing and subsequent incubation for 15 min at 4 °C. The supernatant was recovered by centrifugation and filtered through miracloth. In total, 0.52 volumes of isopropanol were added and the mixture was allowed to stand for 20 min at room temperature, followed by centrifugation at 11,000 x g for 20 min at room temperature. The pellet was redissolved in 30 mL TE 10/50, 100 units of RNAse A were added and allowed to digest 2 h at 37 °C. Solid KCl was added to a final concentration of 2.0 M and the volume was adjusted to 35–40 mL. The sample was centrifuged and the super-natant loaded onto a 50 mL superloop. This was injected into a 550 mL sepharose 6 XK 50/30 column and the pure plasmid was collected in the dead volume. The plasmid was precipitated with 0.5 volumes of isopropanol and redissolved in TE 10/0.1.

An aliquot of 75–85 pmol of plasmid DNA was buffer exchanged to $H_2O$ and mixed with CutSmart buffer (NEB) and water to a final volume of 200 μL. Fifty units of BsaI-HF and 50 units of DraIII-HF were added to digest for 8–10 h, then another 20 units of each enzyme was added to get the digestion to completion (additional 20 units were added if not complete). Sixty units of EcoRV-HF were added and digestion was continued for 6–10 h (Supplementary Fig. 1f–i). Two rounds of stepwise PEG precipitation were performed to separate the excised fragment of interest from the plasmid backbone fragments using concentrations of PEG from 7.0 to 8.5% (Supplementary Fig. 1j, k). After two rounds, a final cleanup step was done using a Zymo Clean and Concentrator 100 column.

**Preparation of fluorescently labeled DNA fragments**. An aliquot of 5–10 nmol of oligonucleotide at a concentration of ~1 mM, washed by ethanol precipitation, was diluted with 25 μL oligo labeling buffer (0.1 M sodium tetraborate, pH 8.5 (9.25 for TFP ester labeling). A 0.6 μL sample was taken and diluted with 50 μL oligo-nucleotide RP-HPLC solvent A (95% 0.1 M TEAA, 5% ACN). An aliquot of 40 μL of this was injected for analysis by RP-HPLC on an InertSustain 3 μm, 4.6 × 150 mm GL sciences C18 analytical column using a gradient of 0–100% oligonucleotide RP-HPLC solvent B (70% 0.1 M TEAA, 30% ACN) in 20 min. An aliquot of 5 μL of 5 mM NHS-ester dye in DMSO was added and the reaction allowed to proceed 4–8 h at room temperature. The progression of the reaction was monitored by RP-HPLC. Further, dye was added if required, until >50% oligonucleotide was labeled. The oligonucleotide was precipitated twice with ethanol to remove residual dye. It was redissolved in 30 μL MQ $H_2O$ and diluted with 70 μL oligo RP-HPLC solvent A. Labeled oligonucleotides were purified by RP-HPLC using the same gradient and column as above and collected manually followed by ethanol precipitation. The purified labeled oligonucleotide was redissolved in MQ $H_2O$ to give a concentra-tion of 2.5 μM (Supplementary Fig. 2a–i).

Labeled PCR segments were generated by mixing Thermopol (1x), template (0.02 ng μL$^{-1}$), forward primer (0.250 μM), reverse primer (0.250 μM), and dNTPs (0.2 mM each) with water in $N \times 50$ μL to the final concentrations given in the parentheses. $N \times 1.25$ units Taq DNA polymerase was added, the solution was gently mixed followed by aliquoting 50 μL into each of N PCR tubes. Thermocycling was done with 12 s initial denaturation at 94 °C followed by 30 cycles of 12 s denaturation at 94 °C, 12 s annealing at 60–65 °C, and 12 s extension at 72 °C. Final extension was done for 12 s at 72 °C. PCR product from the N tubes were pooled and stored in the freezer.

An aliquot of 450–500 μL of PCR product was purified with 3x QIAquick PCR purification columns according to the manufacturer's protocol. Following elution, the DNA was ethanol precipitated and redissolved in ~100 μL MQ $H_2O$. PCR-generated pieces were digested by mixing 75–85 pmol of each piece in 200 μL with 10x CutSmart to a final concentration of 1x and a sample taken. The pieces were digested as done for the recombinant pieces. Samples were taken and analyzed on a 2% agarose gel alongside the undigested pieces (Supplementary Fig. 2j). The digestion reactions were purified with QIAquick PCR purification columns and the concentration was determined by UV spectroscopy.

**Convergent DNA ligations for 12 × 601 arrays**. An aliquot of 30–60 pmol of each DNA piece was used for large-scale ligation to generate the intermediates in combined volumes of 200–400 μL (Supplementary Fig. 3a). P2 was ligated to P1 in 20% excess for 2 h, then P3 was added in 20% excess relative to P2 and ligation allowed to proceed overnight. P4 was ligated to P5 in 20% excess for 2 h, then the biotinylated anchor was added in twofold excess relative to P5 and the ligation allowed to proceed 12–16 h (Supplementary Fig. 3b–d). The pieces were purified by PEG precipitation using a stepwise (0.5% steps) increase in PEG from 7.0 to 8.0% (Supplementary Fig. 3e, f). Pellets were redissolved in 60 μL TE buffer (10 mM Tris, 0.1 mM EDTA, pH 8.0) and redissolved pellets and the final supernatant were analyzed by agarose gel to verify that the pellets at 7.0 and 7.5% typically contained the intermediates separated from the starting pieces. These were pooled and stored for later ligations. An aliquot of 15–35 pmol of the 6 × 601 intermediates were mixed using 5–10% excess P4-P5-anchor in 1x T4 DNA ligase buffer with 600 U of ligase and left to ligate for 10–16 h. The formation of the product was confirmed by agarose gel electrophoresis and purified by stepwise PEG precipitation in the range

5.0–6.0% (Supplementary Fig. 3g, h). The pellets were redissolved in TE(10/0.1) and analyzed by gel electrophoresis to pool the purified double-labeled array DNA.

**Chemically modified histones**. Preparation of H4K$_S$16ac was performed by radical-mediated thiol-ene addition[66]. H4 carrying a K16 to cysteine point muta-tion (K16C) was expressed and purified from inclusion bodies[27]. For the instal-lation of the acetyl-lysine analog, H4K16C was dissolved in 0.2 M acetate buffer, pH 4 to a final concentration of 1 mM. Subsequently, 50 mM N-vinylacetamide, 5 mM VA-044 and 15 mM glutathione were added, and the reaction was incubated at 45 °C for 2 h. The reaction was monitored by HPLC and MS until complete, followed by semi-preparative RP-HPLC purification of the product (Supplemen-tary Fig. 7a, b).

For the synthesis of H3K9me3[27], a peptide corresponding to H3(1–14)K9me3-NHNH$_2$ (carrying a C-terminal hydrazide) was produced by solid phase peptide synthesis. Truncated H3 [H3(Δ1–14)A15C] was expressed as an N-terminal fusion to small ubiquitin like modifier (SUMO) carrying a His6-tag. After a denaturing Ni:NTA affinity purification, the protein was refolded and SUMO was cleaved by SUMO protease, followed by purification of H3(Δ1–14)A15C by RP-HPLC. In a typical ligation reaction, 3 μmol H3(1–14)K9me3-NHNH$_2$ was dissolved in ligation buffer (200 mM phosphate pH 3, 6 M GdmCl) at −10 °C. NaNO$_2$ was added dropwise to a final concentration of 15 mM. The reaction was subsequently incubated at −20 °C for 20 min. H3(Δ1–14)A15C was dissolved in ligation buffer (200 mM phosphate pH 8, 6 M GdmCl, 300 mM mercaptophenyl acetic acid (MPAA)) and added to the peptide. The pH was adjusted to 7.5 and after completion of the reaction (as observed by RP-HPLC), the product (H3K9me3A15C) was purified by semi-preparative RP-HPLC. H3K9me3A15C was finally dissolved in desulfurization buffer (200 M phosphate pH 6.5, 6 M GdmCl, 250 mM tris(2-carboxyethyl)phosphine (TCEP)). Glutathione (40 mM) and a radical initiator, VA-044 (20 mM), were added, and the pH was readjusted to 6.5. The reaction mixture was incubated at 42 °C until the protein was completely desulfurized, followed by semi-preparative HPLC purification (Supplementary Fig. 19a, b).

**Chromatin assembly**. Chromatin arrays were reconstituted on a scale of 6.5–30 pmol (based on 601 DNA). 12 × 601 array DNA was mixed with 1.5 molar excess of MMTV buffer DNA, NaCl to a final concentration of 2 M and water, followed by mixing and addition of 2–2.4 molar equivalents of histone octamers, containing either recombinant or chemically prepared modified histones (Supplementary Figs. 7 and 19). The mixture was transferred to a micro-dialysis tube and dialyzed with a linear gradient from TEK2000 (10 mM Tris, 0.1 mM EDTA, 2000 mM KCl) to TEK10 over 16–18 h. The dialysis tube was transferred to 200–600 mL TEK10 for another 1 h of dialysis. The chromatin assembly mixture was taken out of the dialysis tube and centrifuged at 21,000 x g for 10 min followed by transfer of the supernatant to a fresh tube. The concentration and volume of the mixture was determined. Gel analysis was done with 0.25–0.50 pmol of chromatin assembly sample (calculated based on the total 260 nm absorption and the extinction coef-ficient for each nucleosome repeat) mixed to 10 μL with TEK10 and 5–7% sucrose added from a 25% stock. Samples were run in 0.7% agarose gels made with 0.25x TB, using the same as running buffer at 90 V for 90–100 min.

For ensemble FRET analysis, which requires removal of MMTV DNA and nucleosomes, 5–10% of the volume was taken aside for analysis and the remainder was mixed with an equal volume of 6 mM $Mg^{2+}$ for precipitation on ice for 10 min followed by 10 min centrifugation at 21,000 x g. The supernatant was transferred to another tube and the chromatin pellet redissolved in a similar volume of TEK10 as present prior to precipitation. Similar volumes as taken for chromatin assembly analysis were used for subsequent analysis of the recovered chromatin. For ScaI digestion, a similar volume of sample in 1x CutSmart buffer was mixed with 10 units of ScaI-HF followed by digestion for 5–7 h. Samples of chromatin before and after precipitation and after ScaI digestion were analyzed as described above. Gels were visualized in fluorescence channels and then stained with GelRed for visualization of DNA and nucleosome/chromatin bands (Supplementary Fig. 4a–o).

**Ensemble FRET measurements on chromatin**. Chromatin samples isolated after magnesium precipitation were diluted to a final volume of typically 220–250 μL, resulting in a concentration that yields a spectral count of around 90,000–130,000 cps for maximum donor fluorescence emission, prior to chromatin compaction. The sample was then split in 4 × 50 μL volumes. TEK10 and $Mg^{2+}$ from stocks of 10 or 50 mM was added along with TEK10 to a final volume of 55 μL, 5 min prior to measuring. After standing 5 min, the sample was transferred to the fluorescence micro-cuvette for measurement of the spectra (two repeats), followed by mea-surement of the donor anisotropy (two repeats). This was done for all the samples in the range 0–4 mM $Mg^{2+}$ (Supplementary Fig. 5a–l, o). For all measurements, the following settings were used on the fluorescence spectrometer: excitation at 575 nm with 4 nm slit width, and detection over the range of 585–700 nm with 5 nm slit width. For anisotropy measurements, the emission slit width was opened to 10 nm and measurements were performed at 592 nm.

 

**Preparing of flow chambers**. Borosilicate glass slides with two rows of four holes and borosilicate coverslips were sonicated standing upright in glass containers for 20 min in MQ $H_2O$, then in acetone and then in ethanol. They were cleaned in piranha solution (25% v/v $H_2O_2$ and 75% v/v $H_2SO_4$) in the same glass containers for 1 h, followed by washing with MQ $H_2O$ until reaching neutral pH. A 500 mL Erlenmeyer flask was cleaned in the same way. The Erlenmeyer flask, coverslips and slides were all sonicated in acetone for 10 min. A solution of 3% v/v amino-propyltriethylsilane in acetone was prepared in the Erlenmeyer flask and used to immerse the microscopy glass and incubated for 20 min. The aminosilane was disposed, the slides were washed in water and dried with $N_2$. Flow chambers were assembled from one glass slide and one coverslip separated by double sided 0.12 mm tape positioned between each hole in the glass slide. The ends were sealed with epoxy glue and the silanized slides stored under vacuum in the freezer until use.

Silanized glass flow chambers stored in the freezer were allowed to warm for 20–30 min. Then a pipette tip as inlet reservoir and outlet sources were neatly fitted in each of the 2 × 4 holes on each side of the flow chamber and glued in place with epoxy glue. The glue was allowed to solidify for 30–40 min. Subsequently, 350 μL of 0.1 M tetraborate buffer at pH 8.5 was used to dissolve ~1 mg of biotin-mPEG (5000 kDa)-SVA, and 175 μL from this was transferred to 20 mg mPEG(5000 kDa)-SVA to generate a transparent clouding-point solution after 10 s of centrifugation. This was mixed to homogeneity with a pipette and centrifuged again for 10 s before 40–45 μL aliquots were loaded in each of the four channels in the flow chamber. The PEGylation reaction was allowed to continue for the next 2½–4 h, after which the solution was washed out with degassed ultra-pure water.

**smTIRF measurements**. Measurements were carried out with a micro-mirror TIRF system[67] (MadCityLabs) using Coherent Obis Laser lines at 405, 488, 532 and 640 nm, a 100x NA 1.49 Nikon CFI Apochromat TIRF objective (Nikon) as well as an iXon Ultra EMCCD camera (Andor), operated by custom-made Labview (National Instruments) software. For imaging, buffers with/without biomolecules were deposited in the inlet reservoir of microfluidic flow cells and drawn into the chamber with tubing connected from the outlet to a 1 mL syringe operated manually or with a motor-driven syringe pump. For each experiment, the imaging chambers were washed with 200–300 μL T50 (10 mM TrisHCl, pH 8,5, 50 mM NaCl), followed by incubation with 50 μL 0.2 mg mL$^{-1}$ neutravidin for 5 min. This was washed out with another 400–500 μL T50. Then, 0.5–2 μL of chromatin assembly reaction at a concentration of 5–40 ng μL$^{-1}$ was loaded into the chamber while monitoring acceptor emission, to assess chromatin coverage. Chromatin was loaded until reaching 150–400 chromatin arrays in a 25 × 50 μm field of view. Excess chromatin was washed out with T50 followed by exchange to imaging buffer (40 mM KCl, 50 mM Tris, 2 mM Trolox, 2 mM nitrobenzyl alcohol (NBA), 2 mM cyclooctatetraene (COT), 10% glycerol and 3.2% glucose) supplemented with GODCAT (100x stock solution: 165 U mL$^{-1}$ glucose oxidase, 2170 U mL$^{-1}$ cata-lase). For imaging, a programmed sequence was employed to switch the field of view to a new area followed by adjusting the focus. Then the camera was triggered to acquire 1300–2000 frames with 532 nm excitation and 100 ms time resolution followed by a final change to 640 nm excitation. For sequences requiring timed programmed injection, after 5000 ms the pump was triggered (Fig. 1). For experiments with magnesium and HP1α, the mixture with the desired concentration was prepared and loaded into the inlet reservoir followed by injection into the channel and imaging as described above.

From acquired movies, the background was extracted in ImageJ using a rolling ball algorithm. Trace extraction and analysis was performed in custom-written MATLAB software. The donor and the acceptor images were non-isotropically aligned using a transformation matrix generated from 8 to 10 sets of peaks appearing in both the donor and the acceptor channels. Peaks were automatically detected in the initial acceptor image prior to donor excitation and the same peaks were selected in the donor channel. Peaks that were tightly clustered, close to the edges or above a set intensity threshold in either the donor or the acceptor channels indicating aggregation were removed from analysis. The analysis was then limited to the peaks appearing in both the donor and the acceptor channel and these traces were extracted for further analysis.

Traces were selected based on the following criteria: (1) Initial total fluorescence of the donor and the β-corrected acceptor of >600 counts over baseline (at 900 EM gain). (2) At least 5 s prior to bleaching of acceptor or donor. Note that for bleaching experiments (Fig. 1f, g or Fig. 7h), the required trace length was raised to 10 s. (3) Single bleaching event for donor or acceptor. (3.a) If acceptor bleaches first; leads to anti-correlated increase in donor to same total fluorescence level as prior to bleaching. (3.b) If donor bleaches first, the acceptor dye must still be fluorescent when directly probed at the end of the experiment. (4) Bleaching of the donor dye during the 120 s of acquisition to allow an unambiguous determination of background levels. See Supplementary Fig. 6 for a graphical representation.

**MFD sample cell preparation**. 24 × 40 × 1.5 mm coverslips were silanized as described above for the cleaning and passivation to generate the microfluidic channels. Two silicon gaskets were cut out with a scalpel and placed on top of a coverslip. An aliquot of 20 mg mPEG(5000 kDa)-SVA was suspended in 175 μL 0.1 M tetraborate buffer at pH 8.5. The mPEG-SVA suspension was centrifuged at 13,300 × g for 10 s and pipetted up and down before distributing approximately 40 μL in each silicon gasket on a coverslip to PEGylate. The PEGylation reaction

was allowed to proceed for the next 1–2 h before the solution was washed away by first removing the mPEG(5000 kDa)-SVA solution and then washing three times with MQ $H_2O$. For one of the washes, the water was allowed to stay in the gasket for 5 min before removing it. The gaskets were then filled with measurement buffer, and stayed like this until usage.

**MFD measurement procedures**. Chromatin fibers with the FRET pair Alexa488/647: MFD measurements with pulsed interleaved excitation (PIE) were essentially performed as shown in ref. [68] employing a confocal epi-illuminated setup based on an Olympus IX70 inverted microscope. In PIE measurements, donor and acceptor are sequentially excited by rapidly alternating laser pulses. MFD can be performed on both dyes, allowing computation of the donor–acceptor ratio (stoichiometry, S) for each particle. Excitation is achieved using 485 nm and 635 nm pulsed diode lasers (LDH-D-C 485 and LDH-P-C-635B, respectively; both PicoQuant (Berlin, Germany)) operated at 32 MHz and shifted by 15.625 ns (total frequency of both Lasers 64 MHz) focused into the sample solution by a 60×/1.2 NA water immersion objective (UPLAPO 60x, Olympus, Germany). Laser power in the sample was $L_G = 36$ μW and $L_R = 7.5$ μW, respectively. We used the excitation beamsplitter FF550/646 (AHF, Germany) to split laser light and fluorescence. For confocal detection, a 100 μm pinhole was applied for spatial filtering. The fluorescence photon train was divided into its parallel and perpendicular components by a polarizing beamsplitter cube (VISHT11, Gsänger) and then into spectral ranges below and above 595 nm by dichroic detection beamsplitters (595 LPXR, AHF). After separating, the fluorescence signal according to color and polarization, each of the four channels was split again using 50/50 beamsplitters in order to get dead time free filtered FCS curves, resulting in a total of eight detection channels. Photons were detected by eight avalanche photodiodes (green channels: τ-SPAD-100, PicoQuant; red chan-nels: SPCM-AQR-14, Perkin Elmer). Additionally, green (HQ 520/35 nm for Alexa488) and red (HQ 720/150 nm for Alexa647) bandpass filters (AHF, Ger-many) in front of the detectors ensured that only fluorescence from the acceptor and donor molecules were registered, while residual laser light and Raman scat-tering from the solvent were blocked. The detector outputs were recorded by a TCSPC module (HydraHarp 400, PicoQuant) and stored on a PC. Data were taken for at least 90 min per sample. Bursts of fluorescence photons are distinguished from the background of 0.5–1 kHz by applying certain threshold intensity cri-teria[68]. For analysis, several parameters, including fluorescence lifetime, anisotropy, and FRET efficiency, were computed per burst to classify the molecules according to multidimensional relations between these parameters. For MFD measurements at SMD conditions, assembled chromatin was diluted to a concentration of approximately 50 pM (1–100 μL from assembly stock solution) in measurement buffer (40 mM KCl, 50 mM Tris and 10% v/v glycerol, pH ~7.2) containing the desired amount of magnesium. This was then deposited into the silicon gaskets on a passivated coverslip that had been washed with the same measurement buffer prior to deposition of the sample.

Chromatin fibers with the FRET pair Alexa568/647: MFD measurements were performed with one color excitation using a 530 nm amplified pulsed diode laser (LDH-FA 530B, PicoQuant (Berlin, Germany)) with a repetition rate of 64 MHz. The rest of the setup was identical except the customized dichroic beamsplitters (excitation beamsplitter F68-532m zt532/640/NIR rpc (AHF, Germany), dichroic detection beamsplitters F48-642, T640LPXR (AHF) and bandpass filters (HQ 595/50 (AHF)) for the new donor Alexa568 and adapted bandpass filters (HQ 730/140, (AHF)) for the acceptor Alexa647.

**Dynamic structural biology analysis**. All procedures (11 steps) are outlined in Supplementary Fig. 9 and described in detail in the Supplementary Note. Long timescale dynamics were analyzed by smTIRF (Figs. 1–2 and step 1). Short time-scale dynamics were detected in MFD plots (Fig. 3 and step 2). The FRET efficiency levels corresponding to the chromatin structural states were determined by sub-ensemble fluorescence lifetime measurements (step 3 and Supplementary Figs. 10) and dynamic PDA of signal intensities (step 7 and Supplementary Figs. 16–18). Dynamics were analyzed by burst-ID FCS analysis (step 4 and Supplementary Fig. 11). Contributions from photobleaching and blinking were analyzed (step 5 and Supplementary Fig. 14). Kinetic models consistent with the analysis from steps 1–5 were formulated (step 6, Fig. 5 and Supplementary Fig. 15), and used for fitting using dynamic PDA (step 7 and Supplementary Figs. 16–18). Obtained kinetic and structural models were validated (step 8). Uncertainties in the measured distances were evaluated (step 9) and structural models of compact (step 9, Supplementary Fig. 12) and open chromatin fibers (step 10, Supplementary Fig. 13) were pro-duced. Finally, models were validated to produce a global structural and kinetic model (step 11).

**Code availability**. All custom-made computer code is available upon request from the corresponding authors.

**Data availability**. The smTIRF data sets have been deposited at www.zenodo.org under the accession codes 1040772, 1069675, and 1069677. All other data sup-porting these findings are available from the corresponding authors on reasonable request.

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

## Acknowledgements

We thank Jun-ichi Nakayama for the CK2 expression plasmid, Nicolas Sambiagio for assistance with sample preparations. We thank Manuel M. Müller, Jeffrey C. Hansen, Wilma K. Olson, and Nicolas Clauvelin for stimulating discussions during the initial phase of this research. This work was supported by the Sandoz Family Foundation, the Swiss National Science Foundation (Grant 31003A_173169), the European Research Council through the Consolidator Grant 2017 chromo-SUMMIT (724022) and EPFL (B. F.), the Boehringer Ingelheim Foundation (S.K.), and the European Research Council through the Advanced Grant 2014 hybridFRET (671208) to C.A.M.S.

## Author contributions

B.F. coordinated the project. B.F., S.K. and C.A.M.S. conceived and designed the studies. B.F. supervised chromatin synthesis and reconstitution, and smTIRF studies. C.A.M.S. supervised the confocal smFRET studies and quantitative FRET analysis. S.K. synthesized labeled chromatin fibers, modified histones, and performed smTIRF experiments. I.B. produced phosphorylated HP1α and performed TIRF experiments. L.C.B. produced synthetic histones. S.F., O.D., S.K. and H.V. performed and analyzed confocal FRET experiments. Kinetic modeling was performed by S.F., C.A.M.S. and B.F. PDA was performed by S.F. G.A. performed coarse-grained simulations. G.A. and B.F. generated chromatin models. O.D. and M.D. performed FPS analysis with chromatin models. All authors were involved in data analysis and interpretation. All coauthors wrote the manuscript.

## Additional information

**Competing interests:** The authors declare no competing financial interests.

