## [Peer Review File · Nature Communications]

Reviewers' comments:

Reviewer #1 (Remarks to the Author):

This paper reports that in a dodecamer nucleosome array with 30 bp linkers tetranucleosome stacks can form and their internucleosomal interactions are heterogeneous and dynamic up to 4 mM Mg on a ms and shorter timescale. It also reports that heterochromatin protein 1-alpha (Hp1a) compacts the nucleosome stacks where the internucleosomal interactions still remain dynamic. For the measurements, the authors used smFRET on both surface-immobilized and solution-born nucleosome arrays. Dynamic internucleosomal interactions in an array format at this time resolution has never been reported up to my knowledge, which makes this paper worth publication. I have some concerns, however, as following.

The paper overall lacks information on the statistics of the measurements and results. According to the literature, with 30 bp linkers, one may expect that a nucleosome array would adopt both one-start and two-start helices, which is not the case here. To reconcile this discrepancy, the authors need to clearly state that the vast majority of the arrays they observed display the FRET signals shown in figures 1, 2, and 7. If this is not the case, they need to report what portions of their arrays form tetranucleosomes and state that their results/conclusions apply only when an array forms a tetranucleosome. To further support this issue, the FRET histograms in figure 2 do not match those deduced from the histograms in figure 3, implying that the histograms in figure 2 might be constructed from selected arrays. If so, what were the selection criteria? How big was the sample heterogeneity? Depending on the response to this comment, the significance of this paper could be compromised.

Along the same line, how large a portion of the arrays shows compaction induced by Hp1a as shown in figure 7g/h? How confident are the changes in the FRET histograms and correlation decay times (fig. 7c-f)? Errors of the measurements?

The authors also need to include errors in figures 5/6. According to the very rough fittings shown in figures 3 and 5, the errors in the timescales shown in figures 5/6 must be very large and thus they can considerably weaken the significance of the results.

The kinetic model shown in figure 6 is better than the one in figure 5, although both are only weakly supported by the data. I do not have any issue with the authors proposing this model in figure 6, but in figure 5 I suggest adding at least D' state on panels d-f. Without D', the whole analysis up to this point does not make much sense because of the drastically shorter timescale of B/D equilibrium than that of C/D equilibrium despite the higher compaction of B than that of C (D is an open state).

Reviewer #2 (Remarks to the Author):

The structure of chromatin fibers has been highly debated for decades, and even after the publication of a tetra-nucleosome X-ray structure and an EM reconstruction of a folded fiber there are many open questions, in particular about its dynamics. Here, Kilic and coworkers report detailed FRET measurements on reconstructed 167 NRL fibers that address these questions. By engineering 3 different FRET pairs positions in a 12 nucleosome array they could probe distances between particular points in selected nucleosomes and monitor conformational changes. Two different microscopy approaches, TIRF and confocal MFD, allow for resolving changes in FRET at

timescales between 10's of sub-milliseconds to seconds. Two different dye pairs allowed for a large distance scale range. The data are analyzed in a comprehensive framework that combines both measurement modes, includes detailed structural models and photo physical properties of the dyes. The effects of Mg²⁺, two histone modifications, and HP1a are quantified and integrated in the analysis.

The major finding is that the nucleosome-nucleosome interactions are highly dynamic on the sub-millisecond to 10's of milliseconds. The distances of the FRET probes and their dynamics are consistent with the formation of and exchange between two registers of tetra-nucleosomes. Unwrapping of nucleosomal DNA is limited and compaction of the fiber can be induced by Mg²⁺, and HP1a in combination with H3K9Me3. H4K16Ac, as tested by a close chemical analogue, reduces nucleosome stacking. In all cases the nucleosomes are highly dynamic, though a fraction of the fibers does not exhibit changes in FRET at the 10's of milliseconds time scale.

I truly enjoyed reading this manuscript. It addresses an important topic, and the biochemistry, single-molecule fluorescence experiments and data analysis are all major advances in the field. Production of the FRET labeled 601 arrays is a large effort and is performed and described elegantly. The combination of TIRF and FCS is truly synergistic. The comprehensive data analysis, combining thermodynamic kinetic models, structural models and photophysical models is a laudable effort that is highly beneficial for such complex experiments. The results are well described and discussed and the supplementary data provide sufficient detail that is imperative for proper understanding and reproduction of this work. The conclusions are in line with previous findings but were never presented in such an all-inclusive way. I can therefore fully recommend this manuscript for publication in Nature communications after addressing the minor points below.

Because of the complexity of the data analysis, there are a couple of issues that I cannot fully oversee:

- 1) In figure 3d, in row DA1 there is not a clear population of states A, B and C. However, in figure 1d it appears that 32-60% of the fibers should be in a high FRET state. How can this be related?
- 2) On page 7: 'Two dynamic FRET-lines were required to describe the data (dark and bright blue lines, Fig. 3d), indicating two subpopulations of dynamic chromatin fibers which are distinct within the observation time of ~ 10 ms.' It is not clear that two dynamic lines are necessary to describe the data in these figures. For DA2, it seems that the distribution is better described by a single intermediate dynamic line. In Fig 3e it looks like the dark blue dynamic curve and de static red curve almost overlap. In figures 3d it seems that the light blue line does not capture any subpopulation, except for the low FRET state that is also captured by the two other lines.
- 3) On page 7: 'for all vantage points DA1-3 we observed compact chromatin fibers (EFRET > 0.8) in rapid exchange with extended structures (Fig. 3d)'. However, in Fig 3d for DA1, there are hardly any data points exceeding 0.8.

Are these apparent misfits a result of the global constraints that are applied to the fit? If so, I would recommend a discussion on were the model cannot cover the data properly. Given the many data points and constraints and the complexity of the structure, I would not be surprised that the model cannot cover the entire dataset, but ignoring these issues oversimplifies the problem.

Related, is it possible to extract numbers on the accuracy of the fitted parameters, for example by bootstrapping?

Figs 5g, h and i report the distribution of the static populations. It would be informative to also report the distribution in the dynamic populations as a function of [Mg²⁺] and to show the ratio between dynamic and static populations.

It appears that the static population is dominated by structures that do not have FRET in any of the DA pairs. Could this be due to partial dissociation of a fraction of the fibers? Does this population have a larger diffusion time? Does it increase during the measurements? Is there any

indication of fiber degradation due to dilution?

Page 17: '\... acceptor of > 600.' Please provide units.

Page 6, supplemental data: Probably PEG-SVA was meant instead of PEG.

Reviewer #3 (Remarks to the Author):

The paper by Kilic et al aims at increasing our understanding of structure and dynamics of chromatin, by single molecule FRET measurements. This is a careful study employing state of the art smFRET measurements and analysis. The paper will likely be of interest in the field, since knowledge of chromatin structure and dynamics is key to a better understanding of how gene regulation machinery can have access to the DNA in a compact structure. Their findings are in agreement with an organisation of chromatin in that chromatin is organised in tetranucleosomes with two possible interaction registers. Their findings further convincingly show that these two registers interconvert, and that the interconversion involves an open conformation giving access to the DNA. The authors also investigate the role of the protein HP1 α in chromatin structure and dynamics. While it is clear that much still needs to be learned about the details of this process, the current paper provides very interesting new information.

A few points would benefit from some clarification:

- For the interaction between chromatin and HP1 α the authors invoke the conformer selection model. Without more information about the specificity of the interaction, this is a bit far fetched. How does one imagine HP1 α interacts, at a very detailed level, with chromatin? Does it really form stable complexes that would justify the "conformer selection model"? Compared to what - "induced fit"? If a discussion is really justified at this detail, can they really rule out induced fit ?
- The biological role of the interaction with HP1 α does not become clear.
- The authors state that "a detailed knowledge of the structural states and the timescales of motions within chromatin is thus of critical importance". The authors should stress more how the new data and the obtained model gives us critically more insight into the process of gene regulation etc.
- The authors claim that they "detect that chromatin fibers contain tetranucleosomes as fundamental units". I am not sure this is correct. If I understand the procedures correctly, they chose the positions of the FRET probes with this organisation in mind. So what they can claim is that their data are consistent with this organisation proposed by others.

The paper is overall well and clearly written. I have a few stylistic remarks (use of "enable", "ensure", and "using"). These are in my few grammatical errors but they are unfortunately widespread:

- "enable" should be used in the form "enable somebody to do something", not "enable something"
- "ensure" should be used in the form "ensure something to happen" not "ensure something"
- when using "using", avoid the passive voice

Point-to-point response to reviewer comments

Reviewer #1 (Remarks to the Author):

1. The paper overall lacks information on the statistics of the measurements and results.

Response:

We agree that some of our measurements were lacking rigorous statistical analysis and in other cases the information was only displayed in the SI. We have now added statistical information for all experiments.

In particular, we added:

1. Error bars (s.e.m.) to all E_{FRET} histograms (**Fig. 2d-f** and **Fig. 7c-d**). The number of analyzed traces are given in **Supplementary Table 5**.
2. Error bars (s.e.m.) to all TIRF correlation traces (**Fig. 2g-i** and **Fig. 7e-f**). The number of analyzed traces are given in **Supplementary Table 5**.
3. Error margins (95% fit confidence intervals to all fits of TIRF cross-correlation data (**Fig. 2g-i** and **Fig. 7e-f**) and dynamic TIRF experiments (**Fig. 7h**).
4. Uncertainties in the measured distances have been calculated and are given as a percentage, due to precision of data measurement and analysis and accuracy of the used parameters (e.g. R_0) (**Fig. 4**, for the new calculations see **Supporting Information step 9**, and the new **Supplementary table 7**)
5. Error margins (s.d. of 3 subsampled PDA fits using 70% of the data) to microscopic time constants (**Fig. 5d-f**).
6. Error margins (s.d. of 3 subsampled PDA fits using 70% of the data) to populations as shown in **Fig. 5g-i**.
7. Error margins (s.d.) to displayed time constants in the dynamic register model (**Fig. 6**).

2. According to the literature, with 30 bp linkers, one may expect that a nucleosome array would adopt both one-start and two-start helices, which is not the case here. To reconcile this discrepancy, the authors need to clearly state that the vast majority of the arrays they observed display the FRET signals shown in figures 1, 2, and 7. If this is not the case, they need to report what portions of their arrays form tetranucleosomes and state that their results/conclusions apply only when an array forms a tetranucleosome.

Response:

This is indeed an important point the reviewer raises. All traces corresponding to a predefined set of technical requirements were included into the analyses in **Fig. 2** and **7**.

The requirements were set as follows:

1. Initial total fluorescence of the donor and the β -corrected acceptor of > 600 counts.
2. At least 5 s prior to bleaching of acceptor or donor.
3. Single bleaching event for donor or acceptor. 3.a) if acceptor bleaches first; leads to anticorrelated increase in donor to same total fluorescence level as prior to bleaching. 3.b) if donor bleaches first, the acceptor dye must still be fluorescent when directly probed at the end of the experiment.
4. Bleaching of the donor dye during the 120 s of acquisition to allow an unambiguous determination of background levels.

To further clarify the selection criteria, we added a graphical representation to **Supplementary Fig. 6** (panel g).

Finally, we have clarified this point in the text by stating on page 6:

“Importantly, all traces corresponding to a defined set of selection criteria, e.g. the presence of a donor and an acceptor dye and a minimal trace length in time (see **Materials and Methods**), were included in this analysis.”

Regarding the issue of one-start vs. two start:

We have performed experiments and modeling to demonstrate that one-start chromatin fibers do not contribute to our FRET signal (**SI Fig. 5m,n: models, SI Fig. 5o: Measurements**). We further prepared chromatin fibers with dyes placed in nearest-neighbor nucleosomes (DA1': Alexa488 in nucleosome N5 at position (39), and Alexa647 in nucleosome N6 at position -39). These dyes would be expected to come into close proximity in a solenoid configuration (see model **SI Fig. 5n**). Those arrays do however not exhibit a measurable FRET signal (**SI Fig. 5o**).

In all our experiments, we have however a population of chromatin fibers that exhibit low FRET efficiency, corresponding to distances between 90 – 150 Å. While unlikely, we cannot completely rule out that this heterogeneous ensemble includes one-start configurations that do not involve direct stacking interactions between nearest-neighbor nucleosomes (i.e. as shown in **SI Fig 5m**).

Because of this fact, we can state the following:

1. We can confirm the existence of a large fraction of two-start fibers (75%) in our samples.
2. We can exclude that one-start fiber conformations contribute to the FRET signal
3. We can exclude one-start solenoids that lead to nearest-neighbor nucleosome stacking.
4. We cannot exclude any other chromatin conformation, which will not result in a FRET signal in our labeling schemes.

3. To further support this issue, the FRET histograms in figure 2 do not match those deduced from the histograms in figure 3, implying that the histograms in figure 2 might be constructed from selected arrays. If so, what were the selection criteria?

Response:

We realize that this was not presented in a clear fashion and we have addressed this concern systematically below:

- 3.1. The histograms in Figure 2 were not constructed from selected arrays (as discussed in the response for point 2) but from all smTIRF traces corresponding to a set of pre-defined requirements.

- 3.2. The 2-D histograms in Figure 3d, corresponding to the same type of chromatin arrays as in **Figure 2** (Alexa568/647 labeled) were measured on a MFD-setup lacking the possibility to perform pulsed interleaved excitation (PIE). PIE is however required to efficiently sort-out molecules lacking an acceptor dye. Thus, the observed signal in **Figure 3d** contains contributions from donor-only molecules that exhibit a FRET efficiency $E_{\text{FRET}} = 0$. This information was provided in the SI but not clearly pointed out in the main text. We have thus edited the text in several places to clarify this fact:

- a. On p. 7, we added the information that PIE only applies to excitation at 485 and 635 nm.
- b. On p. 7, we clarified that for Alexa568/647 labeled arrays no PIE was performed: “We performed MFD measurements (exciting at 530 nm, which precluded PIE) for DA1-3, which revealed a complex population distribution involved in dynamic exchange (**Fig. 3d**) not observed in free DNA samples or chromatin fibers only containing donor fluorophores (**Supplementary Fig. 8**). Due to the absence of PIE in those measurements, donor-only labeled chromatin fibers contributed also to the total observed signal.”

This has also been pointed out in the figure legend to **Figure 3**.

- 3.3. Due to the shortcomings of the MFD setup to measure FRET between Alexa568/647 as well as due to the R_0 of these dyes we then performed all measurements using a different labeling scheme (Alexa488/647, **Figure 3e**) where we have the possibilities of PIE. These histograms (**Figures 3e and 5a**)

indeed fit well to the measurements in Fig. 2 (i.e. comparable selection criteria give similar results as observed by completely different measurement techniques and using different dye pairs).

We have added this information to p. 8:

“To delineate the fiber architectures corresponding to these populations, we performed MFD experiments using Alexa Fluor 488 as a FRET donor ($R_0 = 52 \text{ \AA}$), which was used to improve the spatial resolution at shorter distances (**Supplementary Figure 8b**). Importantly, in this configuration the measurements could be performed with PIE which enabled us to solely analyze molecules carrying both a donor and an acceptor dye.”

4. How big was the sample heterogeneity? Depending on the response to this comment, the significance of this paper could be compromised.

Response:

To clarify the question on heterogeneity between individual arrays, we have added error bars (s.e.m.) to all histograms shown in Fig. 2 and 7, following precedent as e.g. in (Wang et al. Nat. Struct. & Mol. Biol. 2016, 23(1), 31).

Regarding the heterogeneity in between individually prepared samples: Experiments were repeated with individually assembled chromatin fibers and the results were consistent in between sample preparations, labeling schemes (different donors) and measurement techniques, TIRF and confocal PIE-MFD (with comparable selection criteria; see response 3.3). Thus we are confident that our experiments are consistent and reproducible.

5. Along the same line, how large a portion of the arrays shows compaction induced by Hp1a as shown in figure 7g/h?

Response:

For figure 1f/g and 7g/h, after employing the general selection criteria (see response to question 2, but with a requirement for a minimal trace-length of 10 s instead of 5 s) only traces that showed a change in FRET upon HP1 injection were included into the analysis. This subset comprised of:

- 65% of all traces (passing the selection criteria) in Fig. 1f
- 74% of all traces in Fig. 1g
- 42% of all traces for HP1a injection in fig. 7h.

The traces that were excluded, in contrast, had the following properties:

- They did not show any FRET at all
- They already exhibited high FRET efficiency at the beginning of the experiment
- They did not exhibit any change in FRET before bleaching

The used percentages of arrays are now clearly stated in the figure legend.

6. How confident are the changes in the FRET histograms and correlation decay times (fig. 7c-f)? Errors of the measurements?

Response:

We have re-analyzed all FRET histograms and added error-bars to the histogram (s.e.m.) to indicate the uncertainty. For the correlation times, we have re-analyzed all cross-correlation (CC) curves, fitting globally all individual CC curves obtained from the individual chromatin fibers. The 95% confidence intervals for the parameters are now indicated in the figure legend.

7. The authors also need to include errors in figures 5/6. According to the very rough fittings shown in figures 3 and 5, the errors in the timescales shown in figures 5/6 must be very large and thus they can considerably weaken the significance of the results.

Response:

To clarify our approach, we have added further Figure panels to **Figure 3 (f and g)** and improved the discussion in the text (two paragraphs on page 9). Moreover, we have added uncertainties to all values in Fig. 3 – 7. In the following we discuss our procedures and the implemented changes in more detail.

First, we want to shortly summarize our analysis procedure. We performed several simultaneous analyses in our experiments, which revealed independent information on chromatin structure and dynamics:

1. MFD plots: This data representation shows a correlation of E_{FRET} and fluorescence lifetime for a visual indication of the involved limiting FRET states (corresponding to concrete molecular conformations) and the involved dynamics (as explained in **Fig. 3b,c**)
2. Time correlated single-photon counting (TCSPC): This analysis directly reveals the individual FRET levels.
3. Fluorescence correlation spectroscopy (FCS): This analysis uncovers the apparent relaxation timescales within the molecular system, however no microscopic time constants or kinetic connectivity can be deduced.
4. Dynamic photon distribution analysis (dynPDA): This analysis links individual FRET states to relaxation processes, and using an appropriate kinetic model, enables the assignment of individual microscopic rate constants.

To demonstrate the robustness of our analysis we have added the panels **Fig. 3f and g**, directly showing FRET substates from fluorescence lifetimes (TCSPC, **Fig. 3f**), and the exchange kinetics from FCS (**Fig. 3g**).

FRET states and FRET-lines (Fig. 3d,e):

The FRET-lines shown in **Fig. 3d,e** are thus not a fit, but rather the result of the procedure 1-4, as described above. The shape of the FRET-lines solely depends on the limiting FRET states, which were consistently resolved in the MFD diagrams, by subensemble TCSPC (**Fig. 3f and Supplementary Fig. 10**) and by PDA (**Fig. 5 and Supplementary Figures 16-18**). As noted above, we have now added uncertainties to the measured distances in **Fig. 4**. The procedure to determine these uncertainties (composed of the accuracy of parameter determination and the precision of the dynPDA analysis) is discussed in the **Supplementary text, step 9** and the determined parameters are summarized in the new **Supplementary table 7**.

Timescales

The model-free FCS analysis shown in **Fig. 3g (and Supplementary Figure 11)** gives already a direct and independent indication of the dynamic timescales and yields similar results compared to the more complex dynPDA. Microscopic relaxation times (inverse sum of forward and backward rate constants) are then obtained from dynPDA fits, shown in **Fig. 5 and in Supplementary Figures 16-18**.

Here, it is important to note that the parameters are a result of a global fit over 24 datasets for each array: 4 time windows (TW) and 6 different Mg^{2+} concentrations (**step 7, page 18 in Supplementary information**). Through the constraints of the global fit, the parameters are well defined.

The global dynPDA analysis involves complex calculations and is thus very computationally expensive, limiting our possibilities of exhaustive bootstrapping. However, to estimate the stability and precision of the fits, we have performed a subsampling analysis of the fitting procedure for all labeling sites 3 times with reduced datasets (70% of data points). The variations in the parameter values were small (<1-5%), indicating the stability of the fits. This is mainly a result of the global analysis procedure, as each parameter is defined by multiple datasets. The standard deviations between these 3 attempts were previously only displayed in the SI. We have now included this variance in **Fig. 5d-f**, and as error bars in **Fig. 5g-i**.

Finally, to give a better picture of the uncertainties, we have calculated standard deviations for processes that were observed from different vantage points (DA1-3). These standard deviations are displayed in **Fig. 6**.

8. The kinetic model shown in figure 6 is better than the one in figure 5, although both are only weakly supported by the data.

Response:

The kinetic models in **Figure 5** and **Figure 6** are formally identical. We changed the presentation in **Fig. 5** to make this similarity more clear. We note that the kinetic connectivity maps shown in **Fig. 5** yield the simplest models that enable a successful fit to the data by dynamic PDA, according to the criteria outlined in the **Supporting information**. See in particular **SI Fig. 15** for a description of the systematic tests of models.

Regarding the level of data support of our model: The general workflow with orthogonal experimental information from distinct fluorescence observables (sub-ensemble fluorescence lifetime analysis (**Supplementary Fig. 10**), model-free fluorescence correlation analysis from DA1-3 (**Supplementary Fig. 11**), dynamic PDA (**Figure 5 d-f, Supplementary Figure 16-18**), and structural analysis (**Supplementary Fig. 12**) corroborated the FRET states for each labeling pair DA1-3. FCS analysis clearly revealed conformational dynamics with at least three relaxation times, thus involving at least four states (states A-D). Thus, we are convinced that, together with the additional lifetime and correlation data, our models are clearly supported by the current multidimensional dataset.

To further strengthen this point, we have included two new figure panels (**Fig. 3f,g**) that show seTCSPC and FCS for DA1 at 1mM Mg²⁺. From this data, global relaxation times and FRET levels can be directly observed, demonstrating molecular dynamics between multiple FRET states.

9. I do not have any issue with the authors proposing this model in figure 6, but in figure 5 I suggest adding at least D' state on panels d-f. Without D', the whole analysis up to this point does not make much sense because of the drastically shorter timescale of B/D equilibrium than that of C/D equilibrium despite the higher compaction of B than that of C (D is an open state).

Response:

We agree with the reviewer that state D must consist of several states, which do not exchange within sub-ms to low-ms timescale. In fact, the chromatin fiber remains “trapped” in a particular register, and a switch of register requires rearrangements within state D (the open chromatin state). For clarification, we have thus redrawn the kinetic connectivity maps in **Fig. 5d-f** to better match the scheme in **Fig. 6**. We further have drawn the line separating registers 1 and 2 through D, to imply that no register exchange can take place within the timescales of the experiment.

In addition, we amended the figure legends in **Fig. 5**:

“Two dynamic equilibria (registers) are observed: Register 1 comprises species A, C, D (as characterized by their inter-dye distance, RDA), exchanging with the indicated relaxation times. Register 2 comprises species B and D in equilibrium. Register exchange within D is not permitted in the model on the investigated timescales.”

Similarly, the figure legend of **Fig. 6** has been amended and now includes the following description:

“Nucleosomes highlighted in blue are labeled and thus observed in the experiment. Numbered states correspond to different chromatin conformations which exhibit the same FRET efficiency for DA1-3 but which can be kinetically differentiated. FRET species A includes conformational states {A1, A2, A3} for which stacked nucleosomes are observed. FRET species B includes all states {B1, B2} corresponding to observation across two neighboring tetranucleosome units. FRET species D include low-FRET states {D1, Dn} including locally unstacked nucleosomes (D1) and the ensemble of open fibers (Dn).”

Reviewer #2 (Remarks to the Author):

Because of the complexity of the data analysis, there are a couple of issues that I cannot fully over-see:

1. In figure 3d, in row DA1 there is not a clear population of states A, B and C. However, in figure 2d it appears that 32-60% of the fibers should be in a high FRET state. How can this be related?

Response:

This was indeed not clearly explained in the manuscript (see also response to **reviewer 1, point 3** (for convenience partially repeated below)):

The data shown in **Fig. 3d** were obtained without PIE due to instrumental constraints. This precluded efficient removal of measurements from donor-only molecules (which exhibit a $E_{\text{FRET}} = 0$). The data in **Fig. 3d** thus contain contribution from donor-only labeled chromatin arrays. In addition to the change in distance sensitivity, this is one of the reasons why the experiments were performed using a different labeling scheme (Alexa 488/647) and using a MFD setup allowing to perform PIE. We have thus edited the text in several places to clarify this fact:

- a. On p. 6, we added the information that PIE only applies to excitation at 485 and 635 nm.
- b. On p.7 we clarified that for Alexa568/647 labeled arrays no PIE was performed: “We performed MFD measurements for DA1-3 (exciting at 530 nm, which precluded PIE), which revealed a complex population distribution involved in dynamic exchange (**Fig. 3d**) not observed in free DNA samples or chromatin fibers containing only donor fluorophores (**Supplementary Fig. 8**). Due to the absence of PIE in those measurements, donor-only labeled chromatin fibers contributed also to the total observed signal.”

This has also been pointed out in the figure legend.

2. On page 7: ‘Two dynamic FRET-lines were required to describe the data (dark and bright blue lines, Fig. 3d), indicating two subpopulations of dynamic chromatin fibers which are distinct within the observation time of ~ 10 ms.’ It is not clear that two dynamic lines are necessary to describe the data in these figures.

Response:

We have clarified this point in the revised manuscript: The dynamic FRET lines are not a direct fit to the experimental data (i.e. the MFD histograms) but rather a result of the whole analysis (especially subensemble time-correlated single-photon counting (seTCSPC), fluorescence correlation spectroscopy (FCS) and photon distribution analysis (PDA)) as outlined in **Supplementary fig. 9**. We have thus added the following sentence to p. 7:

“An iterative multistep workflow (**Supplementary Fig. 9**) allowed us to resolve distinct structural states by their characteristic FRET efficiencies and dynamics. Based on this analysis, the data could only consistently be described by two dynamic FRET-lines (dark and bright blue lines, **Fig. 3d**), indicating two coexisting subpopulations of dynamic chromatin fibers which are distinct within the observation time of ~ 10 ms”

Moreover, we have added two figure panels to **Fig. 3 (f and g)** that directly show seTCSPC and FCS for DA1 at 1 mM Mg²⁺. This corroborates our findings regarding limiting FRET states and 3 dynamic processes. The new figure panels are discussed in two paragraphs on p. 9.

2.1 For DA2, it seems that the distribution is better described by a single intermediate dynamic line.

Response:

Fig. 3d and **e** illustrate that the high-FRET population is distributed between the two FRET lines. We have made a key assumption: That the structure of the underlying limiting states per-se does not

change with Mg^{2+} . What changes are the exchange rate constants and with them the relative populations of molecules occupying the limiting states. This results in shifts of the populations from one FRET-line to another, as clearly seen e.g. in **Fig. 3d** for DA2 or DA3. Within this framework this shift is also the clear indication that two FRET-lines are required to describe the data.

2.2. In Fig 3e it looks like the dark blue dynamic curve and the static red curve almost overlap.

Response:

In **Fig. 3d** (with the FRET pair Alexa568/647) the dynamic contribution of $A \leftrightarrow C \leftrightarrow D$ (dark blue FRET line) is visibly separated. The same process in **3e** is less directly visible by eye in the image, due to the altered distance sensitivity of the Alexa488/647 FRET pair. The two FRET pairs are however still well defined by the combined data. This is due to the fact that FRET lines are defined by independent information from TCSPC, FCS (see also new **Figure panels 3f,g**) and dynPDA. To clarify this fact we added another sentence to the main text, on p. 8, indicating: "Due to the altered distance sensitivity of the Alexa488/647 FRET pair, compact states (*A*, *B* and *C*) are now better resolved, while the dynamic FRET lines fall closer to the static FRET line compared to the data from Alexa568/647."

Finally, the data from Alexa568/647 are consistent with the data from Alexa488/647. Together, this shows that two label-pairs with different R_0 are highly beneficial for a complete understanding of the system.

2.3. In figures 3d it seems that the light blue line does not capture any subpopulation, except for the low FRET state that is also captured by the two other lines.

Response:

The bright blue line (branch $B \leftrightarrow D$) captures a significant fraction of the observed data, in particular at the 4 mM Mg^{2+} concentration (e.g. in DA2 and DA3).

3. On page 7: 'for all vantage points DA1-3 we observed compact chromatin fibers ($E_{FRET} > 0.8$) in rapid exchange with extended structures (Fig. 3d)'. However, in Fig 3d for DA1, there are hardly any data points exceeding 0.8.

Response:

This was indeed not clearly formulated in the original manuscript. Our analysis reveals limiting states with $E_{FRET} > 0.8$ in exchange with open states. This is indicated by the intersection of dynamic and static FRET lines in **Fig. 3d**. For DA1 we do not directly see many individual bursts (i.e. single-molecule detection events) with $E_{FRET} > 0.8$ in the 2D MFD histograms. This is because these very high FRET bursts often contain less than 20 photons emitted by the donor dye. This is the minimum number of photons in the decay histogram to run the fluorescence lifetime fit routine – otherwise the burst is excluded from histogram.

Their existence can however be inferred from the intensity based analysis, i.e. the FRET lines in **Fig. 3d**. As an example, we show another presentation of the same data as E_{FRET} vs burst duration 2D histogram (**Figure R1**). Focusing on the region in the 2D histogram above $E_{FRET} > 0.8$, a population of high FRET bursts can be discerned (green shaded area in **Fig. R1**).

We have thus rephrased the sentence on p.8, indicating that the compact chromatin fibers are inferred from the analysis, rather than directly observed:

"In summary, for all vantage points DA1-3 our analysis revealed compact chromatin fibers ($E_{FRET} > 0.8$) in rapid exchange with extended structures (**Fig. 3d**)."

Figure R1: FRET efficiency versus burst duration.

4. Are these apparent misfits a result of the global constraints that are applied to the fit? If so, I would recommend a discussion on were the model cannot cover the data properly. Given the many data points and constraints and the complexity of the structure, I would not be surprised that the model cannot cover the entire dataset, but ignoring these issues oversimplifies the problem.

Response:

We do not believe that there are significant misfits demonstrated in **Fig. 3d** and **3e**. In contrary, the figures quite consistently illustrate the validity of our models in describing the data.

We however currently lack the ability to directly fit the 2-D MFD histograms due to the extensive computational complexity. Our analysis is thus based on combining the results of several separate analysis procedures, including seTCSPC, burst-ID FCS and dynPDA (see **Supplementary Fig. 9**). DynPDA, in particular, allows us to accurately fit projections of the 2-D histograms into the FRET coordinate. All the fits and residuals are displayed in **Supplementary Fig. 16-18**, which illustrate the good quality of the fits overall.

5. Related, is it possible to extract numbers on the accuracy of the fitted parameters, for example by bootstrapping?

Response:

We have analyzed the procedure, and currently lack the ability to perform an exhaustive bootstrapping analysis, which would require to resample our dataset several hundred-fold. Note that the PDA relies on a global fit of FRET histograms constructed for 4 time windows at 6 different Mg^{2+} concentrations, and includes complex calculations that run for several hours.

However, we have tested the stability of our fits by subsampling. We fitted 3 different reduced datasets (comprising 70% of all datapoints), and have plotted the standard deviations for the given parameters in **Fig. 5**, the text and the **SI Fig. 16-18**.

Moreover, we added Error margins (s.d.) to all displayed time constants in the dynamic register model (**Fig. 6**).

Finally, we have determined the uncertainties in the measured donor-acceptor distances used for structural modeling. This procedure revealed an uncertainty of 2-3% due to the dynPDA (analysis precision) and an 8-9% uncertainty arising from the determination of the Förster radius (parameter accuracy), resulting in a total uncertainty in the measured distances of 8-9%. This analysis is described in **Supplementary text, step 9**, summarized in the new **Supplementary table 7** and reported in **Fig. 4**.

6. Figs 5g, h and i report the distribution of the static populations. It would be informative to also report the distribution in the dynamic populations as a function of $[Mg^{2+}]$ and to show the ratio between dynamic and static populations.

Response:

We realize that the figure 5 was not clearly labeled. The plot in **Fig. 5g,h,i** shows the combined populations of each state (which includes both the dynamic and static populations). Regarding the ratio of static and dynamic fraction: This information is given in the SI (**SI Fig. 16-18**). We have now explicitly referred to this in the figure legend.

We changed the figure and figure legend to clarify these facts:

“Relative combined populations of observed species A-D for DA1 as a function of $[Mg^{2+}]$ (for the individual contributions of static and dynamic molecules see **Supplementary Figs. 16-18**)”

7. It appears that the static population is dominated by structures that do not have FRET in any of the DA pairs. Could this be due to partial dissociation of a fraction of the fibers? Does it increase during the measurements? Is there any indication of fiber degradation due to dilution?

Response:

We cannot exclude that there is a population of fibers that is misfolded or lacks a nucleosome at a critical position generating static molecules in a low-FRET state. We however do not believe that this population is very large, in particular if donor-only molecules can be eliminated by PIE.

Neither in TRIF nor in confocal measurements we see progressive loss of FRET-active molecules over the time of an experiment (which takes around 90 min for MFD experiments). This implies that fibers are stable within this time-frame. The stability of chromatin is however greatly dependent on proper passivation of the flow-cells. In flow-cells that were not passivated using a PEG-brush (but for example using BSA or poly-lysine) we see rapid loss of FRET efficiency over minutes. We thus took great care to optimize surface passivation to a point where the FRET signal was stable over the whole measurement.

We have added two sentences to the methods section (MFD measurements) on p. 18:

“Coverslips used to form the MFD sample cell were passivated with a PEG brush to ensure chromatin stability and prevent sample loss during measurements.”

“Data were taken for at least 90 minutes per sample, during which no loss in overall fluorescence was observed.”

8. Does this population have a larger diffusion time?

Response:

To investigate this we first analyzed the distribution of burst duration time [ms] of the low-FRET population (LF) in comparison to high FRET population (HF) and found that majority of LF bursts demonstrates a burst duration similar to HF bursts (region between vertical red and blue cursors in **Figure R1**).

To determine more exact diffusion time, burst-ID FCS, as displayed in **Fig. 3g** is not directly suitable (there we only measure an apparent diffusion term). Instead, we applied filtered FCS (fFCS) to resolve unperturbed species-specific diffusion times (for reference, see Felekyan et al, ChemPhysChem 13, 1036-1053, 2012; SI ref. 18). The filters are based on specific fluorescence decays of the HF and LF species. Using fFCS, we could analyze species selective autocorrelation curves for species exhibiting LF and HF (**Figure R2**). The analysis of the autocorrelation function shows higher diffusion times for LF species compared to compact HF species, demonstrating that LF species exhibit a more extended conformation, slowing down passage through the confocal volume.

However, it has to be noted that our system is highly dynamic and thus multiple bunching terms contribute to the correlation functions, which makes this analysis sensitive to misfit. We have thus refrained from adding this analysis, and the detailed values, to the manuscript.

Page 17: ‘... acceptor of > 600.’ Please provide units.

Response:

This was changed to: “> 600 counts over baseline (at 900 EM gain).”

Page 6, supplemental data: Probably PEG-SVA was meant instead of PEG.

Response:

This is correct, we have corrected the mistake.

Figure R2: Filtered FCS analysis of LF and HF autocorrelation curves for DA1 at 4mM Mg²⁺. Fits were performed using a diffusion term and 2 bunching terms. The obtained diffusion terms where $t_{diff}(\text{LF}) = 27.6 \text{ ms}$, $t_{diff}(\text{HF}) = 17.4 \text{ ms}$.

Reviewer #3 (Remarks to the Author):

A few points would benefit from some clarification:

1. For the interaction between chromatin and HP1 α the authors invoke the conformer selection model. Without more information about the specificity of the interaction, this is a bit far fetched. How does one imagine HP1 α interacts, at a very detailed level, with chromatin? Does it really form stable complexes that would justify the "conformer selection model"? Compared to what - "induced fit"? If a discussion is really justified at this detail, can they really rule out induced fit ?

Response:

We and others have explored the interaction of HP1 α with chromatin in great details (recent papers include Kilic et al. Nat Commun 2015, Bryan et al. NAR 2017 from our laboratory, or Hiragami-Harnada et al. Nat Comm 2016, Azzaz et al. JBC 2014 and Canzio Mol Cell 2011). HP1 specifically binds H3K9me3 and, most likely, requires multivalent H3K9me3 binding across two nucleosomes to compact chromatin. The interaction is thus quite specific. Nevertheless, invoking a conformational selection mechanism is, as the reviewer correctly points out, speculation at this point. We thus rephrased a sentence in the discussion (p. 17) to read:

“As the chromatin fiber continuously explores a broad conformational space, we speculate that HP1 α molecules capture neighboring nucleosomes through a conformational selection mechanism⁵⁸.”

We further removed the other mention of the conformational selection mechanism.

2. The biological role of the interaction with HP1 α does not become clear.

Response:

HP1 α is thought to contribute to gene silencing by inducing a compact chromatin state with reduced accessibility. We thus added a sentence in the discussion (p. 17):

“This results in an increase in the population of compact states, reducing local chromatin accessibility, rendering chromatin less permissive for transcription and contributing to gene silencing.”

3. The authors state that "a detailed knowledge of the structural states and the timescales of motions within chromatin is thus of critical importance". The authors should stress more how the new data and the obtained model gives us critically more insight into the process of gene regulation etc.

Response:

We agree that this is an important point. We thus expanded a paragraph on the impact of local chromatin dynamics on TF binding, remodeling and transcription in the discussion (p. 15).

“Transcription factors sample the chromatin landscape via non-specific DNA interactions in a combined mechanism of sliding and hopping. Both of these interaction modes require direct access to the DNA within chromatin. Local ...”

In addition, a key prediction from our model is that neighboring TFs might influence their binding behavior by “locking-in” a certain tetranucleosome register (discussed on p. 16). In the near future we will employ our developed technologies to put this hypothesis to the test.

4. The authors claim that the "detect that chromatin fibers contain tetranucleosomes as fundamental units". I am not sure this is correct. If I understand the procedures correctly, they chose the positions of the FRET probes with this organisation in mind. So what they can claim is that their data are consistent with this organisation proposed by others.

Response:

Indeed, we do confirm (rather than newly discover) the role of tetranucleosomes as a fundamental organizational principle. Of note, previously these contacts have however only been indirectly demonstrated in unconstrained fibers in solution (i.e. by crosslinking).

Our main discovery is however that of exchanging tetranucleosome registers determining the short-scale organization of chromatin.

To clarify this point, we rephrased the sentence in the Discussion on p. 14:

“Our data is in agreement with models for the higher-order structure of chromatin fibers containing tetranucleosomes as fundamental units²¹. We further discover significant dynamic heterogeneity because these tetranucleosome units exist in different interaction registers, defining a new supertertiary structure of chromatin.”

5. The paper is overall well and clearly written. I have a few stylistic remarks (use of "enable", "ensure", and "using"). These are i my few grammatical errors but they are unfortunately wide-spread:

- "enable" should be used in the form "enable somebody to do something", not "enable something"
- "ensure" should be used in the form "ensure something to happen" not "ensure something"
- when using "using", avoid the passive voice

Response:

We corrected the text, replacing “enable” with other words (“allow”, “make possible” etc.).

We further replaced “ensure” with “make sure” in one instance.

Finally, we rephrased several sentences containing “using”, either employing active voice, or by replacing “using” with “by”.

REVIEWERS' COMMENTS:

Reviewer #1 (Remarks to the Author):

I find that this revision thoroughly addressed my previous comments. I support its publication.

Reviewer #2 (Remarks to the Author):

The authors addressed all issues adequately and I can recommend this manuscript for publication.

Reviewer #3 (Remarks to the Author):

My comments have been satisfactorily treated.